# POLYRATING: A COST-EFFECTIVE AND BIAS-AWARE RATING SYSTEM FOR LLM EVALUATION

**Jasper Dekoninck, Maximilian Baader, Martin Vechev**
Department of Computer Science
ETH Zurich, Switzerland
{jasper.dekoninck,mbaader,martin.vechev}@inf.ethz.ch

## ABSTRACT

Rating-based human evaluation has become an essential tool to accurately evaluate the impressive performance of large language models (LLMs). However, current rating systems suffer from several important limitations: first, they fail to account for biases that significantly influence evaluation results, second, they require large and expensive preference datasets to obtain accurate ratings, and third, they do not facilitate meaningful comparisons of model ratings across different tasks. To address these issues, we introduce POLYRATING, an expressive and flexible rating system based on maximum a posteriori estimation that enables a more nuanced and thorough analysis of model performance at lower costs. POLYRATING can detect and quantify biases affecting human preferences, ensuring fairer model comparisons. Further, POLYRATING can reduce the cost of human evaluations by up to $41\%$ for new models and up to $77\%$ for new tasks by leveraging existing benchmark scores. Lastly, POLYRATING enables direct comparisons of ratings across different tasks, providing a comprehensive understanding of an LLMs' strengths, weaknesses, and relative performance across different applications. [1]

## 1 INTRODUCTION

Large language models (LLMs) have become powerful tools across a wide range of tasks, sometimes even outperforming human experts (AI@Meta, 2024; Anil et al., 2023; Anthropic, 2024; OpenAI, 2023). To evaluate and compare the performance of LLMs, various benchmarks (Clark et al., 2018; Cobbe et al., 2021; Hendrycks et al., 2021) and evaluation frameworks (Gao et al., 2023; Liang et al., 2022) have been developed. These benchmarks aim to provide a comprehensive evaluation of LLM capabilities across tasks such as code completion, mathematical problem-solving, and multilingual understanding. However, the reliability of these benchmarks to accurately estimate model performance has been questioned due to various concerns about data contamination (Dekoninck et al., 2024; Zhang et al., 2024), errors in ground-truth solutions (Gema et al., 2024), and the discrepancy between benchmarks and real-world performance (Lin et al., 2024).

**Ratings for LLMs** To evaluate LLMs more accurately in real-world scenarios, recent works have made use of rating-based evaluations with human or LLM-based judges (Chiang et al., 2024b; Dubois et al., 2024; Lin et al., 2024). These ratings reflect the relative performance of LLMs on specific tasks and are used to construct leaderboards indicating their real-world performance. As shown in Fig. 1, ratings are derived from preference datasets containing samples consisting of a query $Q$, a response by two models $m^{(0)}$ and $m^{(1)}$, a judge $J$, and a judgment indicating the preferred response $r$. We illustrate human preference datasets for several tasks like code-based (‹/›), mathematical (▦), and Chinese (龍) questions, along with a preference dataset using an LLM-based judge (🖮). Current methods fit each task separately using maximum likelihood estimation (MLE) to obtain ratings $R^i_{\text{task}}$ for each model $m_i$ that predict the judge's preferences as accurately as possible.

**Limitations of Current Rating Systems** However, current rating systems suffer from several critical limitations. First, it is widely recognized that judges are influenced by biases that significantly affect their preferences (Hosking et al., 2023; Wu and Aji, 2023; Shi et al., 2024; Chen et al., 2024).

---

[1]Code is available at https://github.com/eth-sri/polyrating.

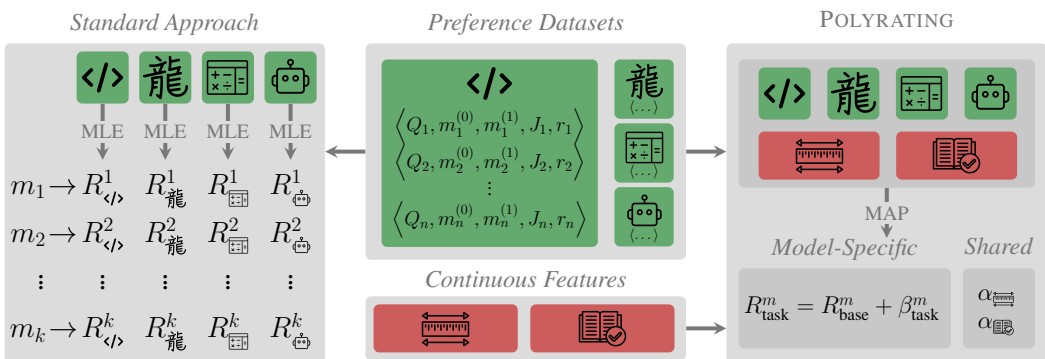

Figure 1: Overview of POLYRATING. Given preference datasets of $n$ samples for $k$ models over various tasks, the standard approach needs to fit separate and independent ratings for each task and cannot leverage continuous features. In contrast, POLYRATING fits a single linear model for all tasks and can leverage continuous features. Attribution in App. A.

Yet, current rating systems are not expressive enough to capture these biases, leading to unfair comparisons of performance. Second, obtaining human annotations is very expensive. However, current systems are sample inefficient and do not take measures to reduce costs. This inefficiency makes it hard for resource-constrained LLM practitioners to use human evaluation for their tasks. Finally, the wide applicability of LLMs requires a comprehensive evaluation system to compare model performance across tasks. Yet, current rating systems suffer from shift-invariance, meaning that rating optimality is preserved when shifting all ratings by an arbitrary constant. For instance, all code-based ratings $R_{\text{</>}}^m$ can be shifted upwards by $40$ points while maintaining optimality, making the ratings on this task much higher than those on other tasks and rendering direct comparisons meaningless.

**This Work: POLYRATING** To address these limitations, we introduce POLYRATING, an expressive rating system designed to model shared continuous features and biases influencing judge preferences. As illustrated in Fig. 1, POLYRATING fits all preference datasets simultaneously using maximum a posteriori (MAP) estimation, i.e. using maximum likelihood estimation with additional priors on all parameters. These priors enable a more robust estimation of model ratings and act as a regularizer to prevent overfitting. Specifically, for each model $m$, a base rating $R_{\text{base}}^m$ is optimized to reflect overall performance across tasks. Task-specific modifiers $\beta_{\text{task}}^m$ are then added to this base rating to derive ratings for individual tasks. Additionally, shared parameters across all models $\alpha_{\text{bias}}$ capture the influence of features such as answer length (⌨) and readability (📖) on preferences.

**Benefits of POLYRATING** POLYRATING addresses previous limitations by design. By modeling shared features, POLYRATING is the first rating system that can quantify biases affecting judge preferences by estimating their impact on model ratings. For instance, we find that answer length bias boosts ratings significantly by $41$ points in the Chatbot Arena (Chiang et al., 2024b). Additionally, POLYRATING improves sample efficiency and reduces the costs of evaluations for new tasks by up to $77\%$ when collecting $10000$ human annotations. POLYRATING can also leverage LLM-based evaluations or traditional benchmarks to obtain ratings for human evaluation, allowing us to reduce its cost by respectively $38\%$ and $41\%$ when collecting $10000$ samples. Furthermore, POLYRATING is not shift-invariant, enabling the construction of a leaderboard that offers detailed insights into each LLM's performance across different tasks, unlike previous approaches. Finally, we provide convergence guarantees and prove the optimality of POLYRATING.

**Main Contributions** In summary, our main contributions are:

- Introducing POLYRATING, a multivariate rating system based on MAP estimation (§3).
- Detecting the influence of several judge biases in human and LLM-based evaluations and for the first time estimating their effect on model ratings (§4.1).
- Demonstrating that POLYRATING improves sample efficiency, reducing the cost of human evaluation by up to $77\%$ for new tasks and by up to $41\%$ for new models (§4.2).
- Providing a multivariate leaderboard using POLYRATING, enabling relative model performance comparisons across tasks (§4.3).

## 2 RATING SYSTEMS

In this section, we introduce the necessary notation to formalize rating systems for LLMs.

**Preference Datasets**    A rating system requires the availability of a preference dataset, which consists of $n$ games that capture the preferences of a judge. In language model evaluation, a game $g$ consists of a user query $Q$, two language models $m_0$ and $m_1$, and a judge $J$. The result $r$ of the game is determined by the judge's preference for one of the completions and is 1 if $m_1$ beats $m_0$, denoted as $m_1 \succ m_0$, and 0 if $m_0 \succ m_1$. Thus, we can represent a game $g$ as a tuple $\langle Q, m_0, m_1, J, r \rangle$.

**Rating System**    For a given set of $k$ models, a rating system assigns a score $\gamma_i \in \mathbb{R}^+$ to model $m_i$, indicating the relative skill of the model on the task. With these scores, the probability that $m_i$ wins against $m_j$ can be computed using the Bradley-Terry model (BT-model) (Bradley and Terry, 1952):

$$P(m_i \succ m_j | \gamma_i, \gamma_j) = \frac{\gamma_i}{\gamma_i + \gamma_j}.$$

In most rating systems the scores are parametrized using the exponential function $\gamma_i = \exp(R_i/400)$ where $R_i$ is the rating of $m_i$ and $400$ is a constant used to scale ratings (Elo, 2008; Glickman, 2002).

**Rating Optimization**    To determine the ratings of the models, a rating system aims to maximize predictive capabilities for the observed outcomes of the games. The maximum likelihood estimate for these observed outcomes in the BT-model can be found by minimizing the logistic loss

$$\mathcal{L}(D, \mathbf{R}) = -\sum_{g \in D} \Big( g_{\mathrm{r}} \log P(g_{m_1} \succ g_{m_0} | \mathbf{R}) + (1 - g_{\mathrm{r}}) \log P(g_{m_0} \succ g_{m_1} | \mathbf{R}) \Big) \tag{1}$$

for a dataset of games $D = (g_1, \ldots, g_n)$ and ratings $\mathbf{R} = (R_1, \ldots, R_k)$. Thus, the optimal ratings can be obtained by computing $\arg\min_{\mathbf{R}} \mathcal{L}(D, \mathbf{R})$. To obtain ratings for specific tasks, the optimization is performed separately for each task on the task-specific dataset $D$.

**Incorporating Draws**    However, the BT-model ignores the possibility of draws in games. Following the approach by the Chatbot Arena (Chiang et al., 2024b), we can generalize the BT-model by setting the outcome $g_r$ equal to $0.5$ for draws to obtain model ratings that can incorporate these draws. Although we explore several alternatives to the BT-model that more explicitly model draws in App. B, we found they did not offer significant advantages in practice.

## 3 POLYRATING

We now introduce POLYRATING, a multivariate rating system specifically designed for language model evaluation. This section first outlines the four design goals for an effective LLM rating system and then explains POLYRATING and how it meets these objectives.

### 3.1 DESIGN GOALS FOR LLM RATING SYSTEM

**1) Quantify Biases**    Both human and LLM-based judges are influenced by biases that affect their preferences (Chen et al., 2024; Hosking et al., 2023; Shi et al., 2024; Wu and Aji, 2023). A robust rating system must capture and quantify these biases to ensure fair comparisons of LLM performance, regardless of the judge. Current rating systems are predominantly univariate and are therefore unable to include the necessary extra parameters to capture these biases. Only AlpacaEval (Dubois et al., 2024) accounts for length bias, where the length of the model answer significantly influences the judge's preference. However, its fitted coefficient is not directly interpretable and does not quantify the bias's influence on ratings.

**2) Leverage Existing Information**    Leveraging existing information can make a rating system more sample efficient and reduce evaluation costs. This information can come from LLM-based evaluations or traditional benchmarks, both of which indicate model performance and should therefore give valuable information that can improve sample efficiency. However, current rating systems start from scratch for each new task and model.

**3) Task Comparability**   Univariate rating systems cannot directly compare ratings across tasks. This is due to shift-invariance, meaning that adding a constant $c \in \mathbb{R}$ to all ratings does not change their optimality. Specifically, the loss of ratings $\mathbf{R}$ on a dataset $D$ is invariant under the transformation $\mathbf{R} \to \mathbf{R} + c$, i.e., $\mathcal{L}(D, \mathbf{R}) = \mathcal{L}(D, \mathbf{R} + c)$. Therefore, if $\mathbf{R}_1$ and $\mathbf{R}_2$ are the optimal ratings for two different tasks, the rating difference for a specific model $R_1^m - R_2^m$ can be shifted by any constant $c$ without losing optimality. This makes it impossible to see how many rating points a model gains or loses from one task to another. A good rating system should eliminate this issue, allowing accurate performance comparisons of the same LLM across different tasks.

**4) Optimality**   Rating systems aim to predict the outcomes of games as accurately as possible. It can be shown that in the limit of infinite data, the univariate rating system presented in §2 obtains the highest possible accuracy. While infinite data is not practically achievable, any new rating system should retain this optimality.

## 3.2   POLYRATING

**Modeling Features**   Features can be continuous, like the length of an answer, or discrete, like the task of the query. We model each feature as a function $f \colon G \times \{0, 1\} \to \mathbb{R}$ that maps a game $g \in G$ and a boolean $i$ to a number quantifying the presence of this feature. The boolean $i$ specifies whether the model for which we want to compute the feature is the first or second model in the game.

**Modeling Ratings**   It is essential to model ratings as a function of these features to capture biases and measure task-specific ratings. For this purpose, we first note that ratings must be game-dependent to incorporate game-dependent features like query task or answer length. Furthermore, judge-specific biases are model-independent and must therefore be measured using parameters that are shared across all models. Finally, it is important that ratings remain interpretable and practical for leaderboards. Therefore, the rating for model $m$ in game $g$ is modeled using the linear model

$$R^m(g) = R_{\text{base}}^m + \sum_{j=1}^{d} \alpha_j f_j(g, [\![g_{m_1} = m]\!]) + \sum_{j=1}^{d'} \beta_j^m f_j'(g, [\![g_{m_1} = m]\!]). \tag{2}$$

Here, $R_{\text{base}}^m \in \mathbb{R}$ is the base rating, $d$ and $d'$ are the number of features in the respective sums, $\alpha_j \in \mathbb{R}$ is the weight for feature $f_j$ and is shared across all models, $\beta_j^m \in \mathbb{R}$ is the model-specific weight for feature $f_j'$, and $[\![\ldots]\!]$ is the indicator function. Importantly, Eq. (2) serves as the key element of POLYRATING that enables it to incorporate all design goals. Indeed, the shared parameters measure the biases in the judge's preferences, addressing the first design goal. Furthermore, to obtain task-specific ratings, we introduce a task-specific feature $f_{\text{task}}'(g, i) = [\![g \in D_{\text{task}}]\!]$, where $D_{\text{task}}$ represents the set of all task-related queries. We then define the task-specific ratings as $R_{\text{task}}^m = R_{\text{base}}^m + \beta_{\text{task}}^m$. While this assumes no additional task-specific features are used, the definition can be easily extended to incorporate them, ensuring that ratings remain adaptable to varying evaluation contexts.

**Optimization Objective**   We perform MAP estimation with a normal prior on the weights $\alpha_j$ and $\beta_j^m$ with mean 0 and deviations $\sigma_j$ and $\sigma_j'$ respectively. This leads to the optimization objective

$$\mathcal{L}_{\text{full}}(D, \mathbf{R}_{\text{base}}, \boldsymbol{\alpha}, \boldsymbol{\beta}) = \sum_{g \in D} \mathcal{L}(\{g\}, R^1(g), \ldots, R^k(g)) + \sum_{j=1}^{d} \frac{(\alpha_j)^2}{2(\sigma_j)^2} + \sum_{m=1}^{k} \sum_{j=1}^{d'} \frac{(\beta_j^m)^2}{2(\sigma_j')^2}, \tag{3}$$

where $\mathbf{R}_{\text{base}}$ is the vector of base ratings, $\boldsymbol{\alpha}$ is the vector of shared weights, $\boldsymbol{\beta}$ is the matrix of model-specific weights, and ratings are computed using Eq. (2). In contrast to the loss function presented in §2, the dataset $D$ can contain games from different tasks.

Using MAP estimation instead of MLE allows POLYRATING to incorporate information from existing tasks through its priors, thereby improving sample efficiency and reducing evaluation costs. Furthermore, evaluations with LLM-based judges can be considered as a distinct task and can therefore be used to improve sample efficiency. Traditional benchmarks can also be reinterpreted as preference datasets, where a question indicates a preference for models that answered correctly over those that did not. If the models are both (in)correct, the judge has no preference. We can therefore leverage all this information, ensuring that POLYRATING satisfies the second design goal.

However, while traditional benchmarks often predict a ranking similar to human preference data, their absolute performance measurements do not always align well with human ratings. For example, an easy benchmark will cluster all models close together because weaker models can answer most questions correctly, making it impossible for stronger models to achieve very high win rates against them. This makes POLYRATING less effective, since the ratings from human preference data are spread further apart than those from these easy benchmarks. To address this issue, we introduce a hyperparameter that rescales the observed win rates of traditional benchmarks. This hyperparameter is optimized on the training data to realign POLYRATING with the human preference data, allowing for a more effective combination of the two. Full details on this parameter are in App. E.

Lastly, POLYRATING removes shift-invariance. Indeed, task-specific ratings $R_{\text{task}}^m = R_{\text{base}}^m + \beta_{\text{task}}^m$ cannot be arbitrarily shifted compared to each other, since the priors on the task-specific weights $\boldsymbol{\beta}$ ensure that $\mathcal{L}_{\text{full}}(D, \mathbf{R}_{\text{base}}, \boldsymbol{\alpha}, \boldsymbol{\beta}) \neq \mathcal{L}_{\text{full}}(D, \mathbf{R}_{\text{base}}, \boldsymbol{\alpha}, \boldsymbol{\beta} + c)$ for any non-zero constant $c$. This enables accurate comparisons of model performance across tasks and fulfills the third design goal.

**Optimization**   In App. D, we show that the optimization objective from Eq. (3) is convex and twice differentiable with respect to $\mathbf{R}_{\text{base}}$, $\boldsymbol{\alpha}$ and $\boldsymbol{\beta}$, allowing us to use standard optimization techniques to optimize the loss with respect to these parameters. Specifically, we use Newton's method for the model-specific parameters and L-BFGS for the shared parameters. Furthermore, we prove in App. D that, under weak assumptions, the obtained ratings converge to the ratings that maximize predictive capabilities and thus fulfill the final design goal.

**Rating Uncertainty**   We compute pivotal intervals using bootstrapping to obtain rating uncertainties (Tibshirani, 1984). Specifically, given the actual estimate of the ratings $\hat{R}$ and $n$ bootstrap estimates $\hat{R}_1, \ldots, \hat{R}_n$, the pivotal interval for a confidence $\alpha$ is $[\hat{R} - \hat{R}_{(1-\alpha/2)}, \hat{R} - \hat{R}_{(\alpha/2)}]$, where $\hat{R}_{(1-\alpha/2)}$ and $\hat{R}_{(\alpha/2)}$ are respectively the $1 - \alpha/2$ and $\alpha/2$ quantiles of the bootstrap estimates. For brevity, we report the $2\sigma$ confidence intervals obtained using bootstrapping in §4, with detailed pivotal intervals reported in App. F.

## 4   EVALUATION

We perform a series of experiments with POLYRATING that showcase its ability to quantify the influence of biases on the ratings of the models (§4.1), its improved sample efficiency for various use-cases (§4.2), and its ability to obtain reliable and comparable multivariate leaderboards (§4.3).

### 4.1   BIAS DETECTION

We use POLYRATING to quantify the influence of biases in both human and LLM-based evaluation using a public subset of the Chatbot Arena (Chiang et al., 2024a) and Wildbench (Lin et al., 2024). This enables an accurate estimation of the effects of these biases on model ratings and allows us to compare the influence of these biases between human and LLM-based judges.

**Biases**   We briefly explain the measured biases and refer to App. E for a full overview of all biases along with their functional form. We include the well-known length bias (Dubois et al., 2024; Singhal et al., 2023), which measures bias with respect to the length of the completion, and position bias (Shi et al., 2024; Wang et al., 2023), which measures bias with respect to the order of the models in the game. We further use classifiers (Babakov et al., 2023; Camacho-collados et al., 2022) to evaluate the influence of formality and sentiment of the model's output. Finally, we compute the repetitiveness of the answer by measuring the number of unique tokens in the completion and check the influence of the readability of an answer using the Flesch Reading Ease score (Kincaid et al., 1975). We model all of these biases as shared features when fitting POLYRATING, allowing us to accurately estimate their influence on the resulting ratings.

**Results**   Table 1 shows the effects of these biases on both human and LLM-based judges. We present both the coefficient $\alpha_{\text{bias}}$ and the average influence this coefficient has on model ratings for given queries, i.e. $\mathbb{E}_g(\alpha_{\text{bias}} \cdot |f_{\text{bias}}(g, 0) - f_{\text{bias}}(g, 1)|)$. To put these numbers into context, the difference between the first and the tenth best models in the Chatbot Arena is 50 rating points.

Table 1: Fitted coefficients for the biases and their average influence on model ratings for both human and LLM-based evaluation. The functional form of $f_{\text{bias}}$ used for each bias can be found in App. E. The influence is computed as $\mathbb{E}_g(\alpha_{\text{bias}} \cdot |f_{\text{bias}}(g, 0) - f_{\text{bias}}(g, 1)|)$.

<table>
<tr><td colspan="3" align="center">(a) Human Evaluation</td><td colspan="3" align="center">(b) LLM-based Evaluation</td></tr>
<tr><td>Bias</td><td>Coefficient ($\alpha$)</td><td>Influence ($\mathbb{E}$)</td><td>Bias</td><td>Coefficient ($\alpha$)</td><td>Influence ($\mathbb{E}$)</td></tr>
<tr><td>Length</td><td>$130.74_{\pm 7.3}$</td><td>$40.84_{\pm 2.3}$</td><td>Length</td><td>$251.87_{\pm 7.2}$</td><td>$48.48_{\pm 1.4}$</td></tr>
<tr><td>Position</td><td>$2.70_{\pm 2.4}$</td><td>$2.70_{\pm 2.4}$</td><td>Position</td><td>$37.53_{\pm 1.1}$</td><td>$37.53_{\pm 1.1}$</td></tr>
<tr><td>Formality</td><td>$-119.89_{\pm 11.3}$</td><td>$-15.17_{\pm 1.4}$</td><td>Formality</td><td>$-37.56_{\pm 6.9}$</td><td>$-4.31_{\pm 0.8}$</td></tr>
<tr><td>Sentiment</td><td>$57.42_{\pm 11.1}$</td><td>$7.90_{\pm 1.5}$</td><td>Sentiment</td><td>$4.31_{\pm 6.5}$</td><td>$0.43_{\pm 0.7}$</td></tr>
<tr><td>Repetitiveness</td><td>$-22.10_{\pm 8.6}$</td><td>$-4.64_{\pm 1.8}$</td><td>Repetitiveness</td><td>$75.04_{\pm 8.3}$</td><td>$9.12_{\pm 1.0}$</td></tr>
<tr><td>Readability</td><td>$72.93_{\pm 11.9}$</td><td>$10.75_{\pm 1.8}$</td><td>Readability</td><td>$-32.56_{\pm 8.0}$</td><td>$-3.92_{\pm 1.0}$</td></tr>
</table>

**Discussion for Length and Position Bias**  We recover prior results on length and position bias for both human and LLM-based judges (Dubois et al., 2024; Singhal et al., 2023; Shi et al., 2024; Wang et al., 2023). Length bias is significant in both paradigms, though more so for LLM-based judges. This explains why length-controlling techniques achieve higher correlation with human judges (Dubois et al., 2024). In contrast, position bias is only significant for LLM-based judges. Furthermore, in contrast to prior work, we can now estimate the effects of these biases on the ratings of the models, with position bias gaining a model around 38 rating points when using an LLM-based judge and length bias gaining 41 and 48 rating points for human and LLM-based judges respectively.

**Discussion for Other Biases**  The other biases reveal interesting patterns. First, all biases differ significantly between human and LLM-based judges. For instance, while readability increases the rating of a model by 11 points for human judges, it decreases the rating by 4 points for LLM-based judges. This indicates that LLM-based judges prefer denser text, while human judges prefer more readable text. Furthermore, we find that sentiment and formality have a significant influence on human judges, gaining models 8 and 15 rating points respectively. In contrast, LLM-based judges are indifferent to sentiment and not influenced as much by formality. Lastly, we find that repetitiveness decreases rating on average by 4 rating points for human judges, while for LLM-based judges, it increases the rating by 9 points.

## 4.2 IMPROVED SAMPLE EFFICIENCY

We demonstrate that POLYRATING is more sample-efficient compared to traditional univariate approaches and therefore reduces the cost of human evaluation. In App. C, we show that a slight adjustment of POLYRATING additionally enables an improved sample efficiency when evaluating new model versions.

**Dataset**  We use the full Chatbot Arena dataset (Chiang et al., 2024b), which contains over one million questions across various tasks. Each question is answered by two models and judged by the human that posed the question. We use the tasks that contain Chinese, code-based, and hard questions for this experiment and refer for a full description of these tasks to App. E.

**New Task**  We first showcase the improved sample efficiency when obtaining ratings for a new task. For this purpose, we vary the number of available questions from the task and compute the logistic loss with respect to a hidden test set. In this process, and in all further experiments of this subsection, POLYRATING is allowed to use all questions in the dataset that do not belong to the task. Thus, we model the rating of a model $m$ for a game $g$ as $R_{\text{base}}^m + \beta_{\text{task}}^m \cdot [\![g \in D_{\text{task}}]\!]$ where $D_{\text{task}}$ is the dataset of all games from the task. The standard deviation of the prior on $\beta_{\text{task}}^m$ is determined by running cross-validation on the current training set.

Results for the three tasks are shown in Fig. 2. We find that POLYRATING converges much faster than the univariate baseline. Measuring efficiency improvement as the fraction of extra samples the univariate baseline requires to obtain the same loss as POLYRATING when collecting 10000 samples, we find that POLYRATING improves efficiency by 58%, 38%, and 77% for respectively the Chinese, code-based, and hard questions. Thus, POLYRATING can cut the cost of obtaining ratings for new tasks by up to fourfold.

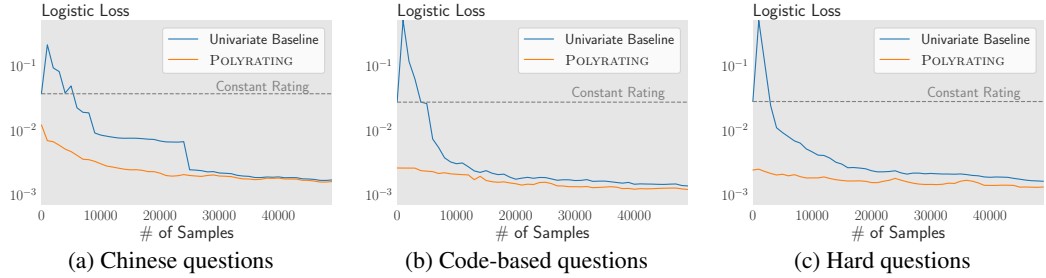

(a) Chinese questions     (b) Code-based questions     (c) Hard questions

Figure 2: Comparison between POLYRATING and univariate baseline for different tasks. The $x$-axis shows the number of samples of the task the rating systems are using. The logistic loss shown is normalized by subtracting the loss of the best possible rating for that task. The grey horizontal line indicates the loss of a rating system that assigns the same rating to all models.

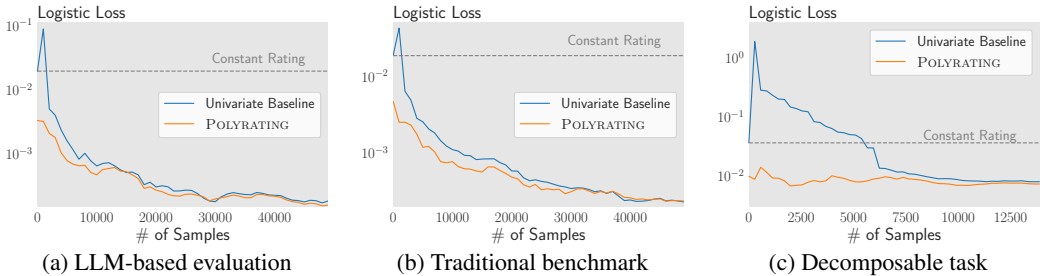

(a) LLM-based evaluation     (b) Traditional benchmark     (c) Decomposable task

Figure 3: Comparison between POLYRATING and the univariate baseline when leveraging information from existing benchmarks. For the left and middle plot, the $x$-axis shows the number of human annotations used. For the right plot, the $x$-axis shows the amount of samples from the Chinese code task. The logistic loss is normalized by subtracting the loss of the best possible rating.

**LLM-based Judge Improves Sample Efficiency** LLM-based preferences are much cheaper to obtain than human preferences. Therefore, LLM-based judges are often used to obtain model ratings, despite being less reliable than human judges. However, POLYRATING can further leverage these ratings to converge faster to the ratings corresponding to human judges. Specifically, we model the rating of a model $m$ for a game $g$ as

$$R^m(g) = R^m_{\text{base}} + \alpha_1 \cdot [\![g \in D_{\text{LLM}}]\!] \cdot \log(\text{length}(g_{y_m})) + \beta^m_1 \cdot [\![g \in D_{\text{LLM}}]\!]$$

where $\text{length}(g_{y_m})$ is the length of the models' completion for the given question. We use the public dataset from Wildbench (Lin et al., 2024) to obtain our LLM-based evaluation. Fig. 3(a) shows the logistic loss on a test set of the Chatbot Arena for a varying amount of human annotations. We find that POLYRATING converges faster to the optimal ratings than the univariate baseline. Specifically, the increase in sample efficiency when collecting 10000 human annotations is 38%.

**Classical Benchmarks Improve Sample Efficiency** Obtaining results from classical benchmarks is even cheaper than LLM-based evaluations. We leverage these benchmarks to increase sample efficiency. To do so, we first convert a benchmark to a preference dataset based on model accuracy as explained in §3. We then model rating as $R^m_{\text{base}} + \beta^m_1 \cdot [\![g \in D_{\text{benchmark}}]\!]$ and apply POLYRATING to increase sample efficiency for human evaluation. We specifically use the MixEval-Hard (Ni et al., 2024) benchmark and show results in Fig. 3(b). We find that POLYRATING significantly increases the sample efficiency. Specifically, when collecting 10000 samples efficiency increases by 41%.

**Decomposable Task** Some tasks can be decomposed into two or more subtasks. If these subtasks are prevalent in the dataset, but their combination is not, POLYRATING can obtain more reliable ratings for the combined task. For example, Chinese code-based questions are rare in the Chatbot Arena, but both Chinese questions and code-based questions are prevalent. For this example, we can model the rating of a model $m$ for a game $g$ as

$$R^m(g) = R^m_{\text{base}} + \beta^m_1 \cdot [\![g \in D_{\text{chinese}}]\!] + \beta^m_2 \cdot [\![g \in D_{\text{code}}]\!] + \beta^m_3 \cdot [\![g \in D_{\text{code}} \cap D_{\text{chinese}}]\!].$$

Fig. 3(c) shows the logistic loss for varying numbers of Chinese code-based questions from the Chatbot Arena. POLYRATING's logistic loss at the start is almost as low as the univariate baseline's at the end, with a sample efficiency improvement of 25% when obtaining 10000 samples.

Table 2: Several models in a multidimensional leaderboard fitted using POLYRATING. The given rank of the models indicates its rank in the complete leaderboard shown in Table 7 in App. F.

| Rank | Model Name | Rating | English | Chinese | Hardness | Code |
|---|---|---|---|---|---|---|
| 1 | gpt-4o-2024-05-13 | $1297_{\pm4.4}$ | $-13_{\pm7.4}$ | $-\ 3_{\pm8.7}$ | $13_{\pm7.5}$ | $15_{\pm8.0}$ |
| 2 | claude-3-5-sonnet-20240620 | $1286_{\pm7.1}$ | $-29_{\pm10.7}$ | $-\ 19_{\pm13.8}$ | $14_{\pm11.0}$ | $44_{\pm12.6}$ |
| 12 | yi-large-preview | $1237_{\pm4.3}$ | $-\ 4_{\pm7.6}$ | $37_{\pm8.5}$ | $17_{\pm7.6}$ | $11_{\pm8.3}$ |
| 26 | llama-3-70b-instruct | $1187_{\pm2.9}$ | $67_{\pm6.0}$ | $-\ 51_{\pm5.4}$ | $-\ 1_{\pm5.6}$ | $-10_{\pm6.1}$ |
| 56 | mixtral-8x7b-instruct-v0.1 | $1114_{\pm4.1}$ | $25_{\pm7.1}$ | $-\ 37_{\pm7.3}$ | $10_{\pm6.8}$ | $-\ 4_{\pm7.6}$ |

Table 3: Several models in the leaderboard fitted using a unidimensional approach where each task is fitted separately. Modifiers for the tasks are shown instead of the fitted rating to make comparison with POLYRATING easier. The complete leaderboard is shown in Table 8 in App. F.

| Rank | Model Name | Rating | English | Chinese | Hardness | Code |
|---|---|---|---|---|---|---|
| 1 | gpt-4o-2024-05-13 | $1283_{\pm3.2}$ | $-19_{\pm5.2}$ | $52_{\pm10.4}$ | $5_{\pm6.7}$ | $13_{\pm7.8}$ |
| 2 | claude-3-5-sonnet-20240620 | $1267_{\pm4.8}$ | $-23_{\pm8.1}$ | $45_{\pm14.4}$ | $11_{\pm10.1}$ | $35_{\pm11.3}$ |
| 10 | yi-large-preview | $1233_{\pm3.3}$ | $-19_{\pm5.6}$ | $84_{\pm10.8}$ | $7_{\pm7.1}$ | $11_{\pm8.3}$ |
| 17 | llama-3-70b-instruct | $1202_{\pm2.5}$ | $22_{\pm4.1}$ | $-\ 32_{\pm8.0}$ | $-\ 5_{\pm5.3}$ | $0_{\pm6.3}$ |
| 53 | mixtral-8x7b-instruct-v0.1 | $1114_{\pm0.0}$ | $0_{\pm0.0}$ | $0_{\pm0.0}$ | $0_{\pm0.0}$ | $0_{\pm0.0}$ |

### 4.3 MULTIVARIATE LEADERBOARD

We now compare separately fitted univariate leaderboards with a multivariate leaderboard fitted using POLYRATING. Since the univariate approach is shift-invariant, we need to fix the shifting constant for each task. We follow the approach of the Chatbot Arena and set the constant by fixing the rating of Mixtral-8x7b-Instruct-v0.1 to $1114$ for all tasks. We will show that this approach fails to provide comparable ratings for the models in the leaderboard.

**Results** Table 2 and Table 3 show the ratings of several models in the leaderboard fitted using POLYRATING and a univariate approach respectively. For a full overview of all models, we refer to App. F. By examining the modifiers computed by POLYRATING, we immediately see the downside of the univariate approach. The Mixtral model performs significantly worse, resp. better, on the Chinese, resp. English, task compared to its base rating. Therefore, fixing the shifting constant using Mixtral results in significant ratings shifts for these tasks making cross-task comparisons impossible.

This effect is most apparent in the Chinese task. 97 of the 114 models included in the dataset gain rating for this task in the univariate approach, even though most models were not specifically trained for the Chinese language. In contrast, POLYRATING shows that only 67 models gain rating, and that the only ones to do so significantly are models trained by Chinese model providers, such as Yi, Qwen and GLM.

Inspecting the English task, we find that POLYRATING indicates that top models tend to lose rating for this task, while bad models tend to gain rating. This is in line with expectations as older and worse models were predominantly trained on English data, making them more suitable for this task and thus increase their rating. However, the univariate approach simply shows that more than 100 models lose rating for this task, with no discernable pattern.

## 5 RELATED WORK

**Ratings** Rating systems have been used across various domains, such as sports (Elo, 2008; Glickman, 2002; Shelopugin and Sirotkin, 2023; Sismanis, 2010; Vaz et al., 2012), gaming (Herbrich et al., 2007; Dangauthier et al., 2007), movies (Talattinis and Stephanides, 2022) and recommendation systems (Adomavicius et al., 2005; Chen et al., 2018; Kong et al., 2019). The widely recognized Elo rating system (Elo, 2008) and its extensions such as Glicko (Glickman, 2002) are generic univariate systems based on the BT-model (Bradley and Terry, 1952) that are widely applicable. Furthermore, various rating systems have been developed for specific use cases and areas. For example, Elo++ (Sismanis, 2010) was specifically designed for chess, and TrueSkill (Herbrich et al., 2007; Dangauthier et al., 2007) has been further developed specifically for multiplayer online games.

**Ratings for LLMs**   Preference datasets for LLMs have become common to evaluate model capabilities in areas lacking ground-truth benchmarks. The most popular one is the Chatbot Arena (Chiang et al., 2024b), which contains over one million user queries and evaluates models in various tasks such as code, math, and multilingual understanding. Wildbench (Lin et al., 2024), MT-Bench (Zheng et al., 2023), and AlpacaEval (Dubois et al., 2024) are LLM-based evaluation frameworks that have gained attention. Among these, AlpacaEval is the only one that applies a length-control bias similar to POLYRATING to obtain a higher correlation with human judges. However, this fitted bias is not directly interpretable and is less generic than the approach used in POLYRATING. Additionally, POLYRATING employs priors on various terms to improve sample efficiency and eliminate shift-invariance, which AlpacaEval lacks. Therefore, AlpacaEval cannot be used to obtain any of the benefits of POLYRATING that were presented in §4.

**Multivariate Rating Systems**   Multivariate rating systems have been used before in recommendation systems (Chen et al., 2018; Adomavicius et al., 2005; Kong et al., 2019; Abdi et al., 2021). These developed systems are extensions to the more classical Elo (Elo, 2008) and Glicko (Glickman, 2002) rating systems. However, they are not directly applicable to LLM evaluation, as they do not take into account the specific biases and dependencies that are present in LLM evaluation. Furthermore, the limited numbers of models allow us to build an exact optimization algorithm, unlike in recommendation systems where approximate algorithms are necessary due to the high number of rated players (or products). These approximate algorithms are not suitable for LLM evaluation, as shown in App. B. Furthermore, these systems do not include priors on the ratings, which are crucial for the sample efficiency of POLYRATING.

**Biases in Human and LLM-Based Evaluation**   Several works have examined biases in both human and LLM-based evaluations (Hosking et al., 2023; Clark et al., 2021; Wang et al., 2023; Wu and Aji, 2023; Shi et al., 2024; Chen et al., 2024; Singhal et al., 2023). Typically, these studies introduce biases to model answers to observe their impact on judge preferences (Wu and Aji, 2023; Chen et al., 2024; Singhal et al., 2023; Wang et al., 2023). Additionally, they also investigate bias by asking more specific questions to the judges, rather than simply asking their preference (Hosking et al., 2023; Wu and Aji, 2023). These techniques, however, do not apply to existing datasets and require additional annotations for specifically crafted answers. In contrast, POLYRATING can be directly applied to existing datasets without further annotation.

## 6   LIMITATIONS

We briefly discuss the limitations of POLYRATING. First, while POLYRATING provides a way to measure model strengths and weaknesses, these comparisons are relative to the other models in the leaderboard and do not provide an absolute measure of model performance. For instance, if *all* models in the leaderboard perform well on one task, and poorly on another, the leaderboard will not reflect this absolute weakness. Instead, it will only show weaknesses relative to the average performance of the models. This is a fundamental limitation of rating systems and cannot be solved by any system that works solely based on preference data. To obtain absolute measures of model performance additional data sources, such as traditional benchmarks, are required.

Furthermore, POLYRATING still requires significant manual inspection and tuning since users must determine the modeling parameters and functions that constitute the rating, a process that can be time-consuming. A more automatic discovery of interesting and relevant dimensions, especially for bias detection, would help mitigate this issue.

## 7   CONCLUSION

We introduced POLYRATING, a multivariate rating system specifically designed for language model evaluation. POLYRATING enables a more comprehensive evaluation of LLMs by capturing biases and dependencies on both continuous and categorical features in the evaluation. We demonstrated the existence and influence of several biases, such as length and position bias, and compared these biases between human and LLM-based judges. Furthermore, we showed that POLYRATING can leverage existing data to increase sample efficiency by $41\%$ and reduce the costs of human evaluations for new tasks by up to $77\%$. Finally, we showed that POLYRATING can provide a more reliable performance comparison of the same language model across different tasks by solving the shift-invariance of the ratings across multiple dimensions.

## REPRODUCIBILITY STATEMENT

We have included our code in the supplementary material with instructions how to run and reproduce all the results presented in the paper. Furthermore, App. D contains detailed proofs of all theoretical statements made in the paper, particularly with respect to the optimality of POLYRATING.

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

## A  ATTRIBUTION

We provide attribution for the icons used in Fig. 1 here. The code icon was obtained from flaticon.com and created by Royyan Wijaya. The Chinese icon was obtained from flaticon.com and created by Freepik. The bot icon was obtained from flaticon.com and created by Nuriali. The math icon was obtained from flaticon.com and created by widphic. The length icon was obtained from freepik.com and created by Surang Lineal. Finally, the readability icon was obtained from freepik.com and created by Generic Detailed Outline.

## B  ALTERNATIVE RATING SYSTEMS

This section explores several alternatives to the exponential rating system that solves the MLE of the logistic loss function, as discussed in §2. Specifically, we evaluate two extensions to the BT-model and one alternative inspired by the accuracy metric commonly used in benchmarks. We then compare these alternatives with the exponential rating system in terms of their predictive performance and demonstrate that their added complexity does not result in better predictions.

All models discussed here are compatible with POLYRATING and can be used as substitutes for the MLE-based BT-model used in §4.

**Rao-Kupper Model**   Rao and Kupper (1967) extend the BT-model to explicitly account for the probability of a draw by introducing a parameter $\theta \in \mathbb{R}, \theta \geq 1$:

$$P(i \succ j|\gamma_i, \gamma_j) = \frac{\gamma_i}{\gamma_i + \theta\gamma_j}$$
$$P(j \succ i|\gamma_i, \gamma_j) = \frac{\gamma_j}{\gamma_j + \theta\gamma_i}$$
$$P(i \simeq j|\gamma_i, \gamma_j) = \frac{\gamma_i\gamma_j(\theta^2 - 1)}{(\gamma_j + \theta\gamma_i)(\gamma_i + \theta\gamma_j)}$$

It can be shown that this model follows from the hypothesis that a judge cannot tell the difference between two answers if the quality of the answers is close to each other.

**Davidson-Model**   Davidson (1970) propose a similar modification to include draws, using a parameter $\theta \in \mathbb{R}, \theta \geq 0$:

$$P(i \succ j|\gamma_i, \gamma_j) = \frac{\gamma_i}{\gamma_i + \gamma_j + \theta\sqrt{\gamma_i\gamma_j}}$$
$$P(j \succ i|\gamma_i, \gamma_j) = \frac{\gamma_j}{\gamma_i + \gamma_j + \theta\sqrt{\gamma_i\gamma_j}}$$
$$P(i \simeq j|\gamma_i, \gamma_j) = \frac{\theta\sqrt{\gamma_i\gamma_j}}{\gamma_i + \gamma_j + \theta\sqrt{\gamma_i\gamma_j}}$$

**Accuracy-Based Model**   Both extensions to the BT-model presented above still model ratings using an exponential function. However, for LLMs, it could be beneficial to use ratings directly comparable to standard benchmark accuracies. Benchmarks can be viewed as a series of games where, for a given question $Q$, model $m_1$ defeats model $m_2$ if $m_1$ answers correctly and $m_2$ does not. A draw occurs if both answer correctly or incorrectly, and otherwise $m_2$ wins.

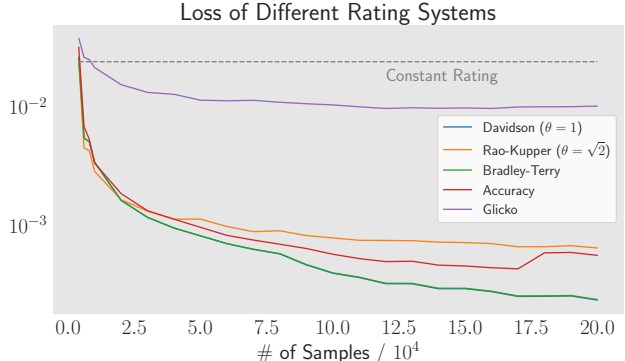

Figure 4: Logistic loss for all four alternatives on the Chatbot Arena dataset for various sizes of the training set.

Let $\text{Acc}_D$ denote the accuracy function on benchmark $D$. If we model draws as $0.5$ points for each model, the win rates can be expressed as:

$$P(i \succ j | m_i, m_j) = \frac{1}{2}\Big(1 + \text{Acc}_D(m_1) - \text{Acc}_D(m_2)\Big)$$
$$P(j \succ i | m_i, m_j) = \frac{1}{2}\Big(1 + \text{Acc}_D(m_2) - \text{Acc}_D(m_1)\Big).$$

To adapt the BT-model to this accuracy-based approach, we modify it as follows:

$$P(i \succ j | R_i, R_j) = \min\left(1, \max\left(0, \frac{1}{2}(1 + R_i - R_j)\right)\right)$$
$$P(j \succ i | R_i, R_j) = \min\left(1, \max\left(0, \frac{1}{2}(1 + R_j - R_i)\right)\right),$$

where the $\min$ and $\max$ functions ensure probabilities remain within the $[0, 1]$ range. Fitting this model on a standard accuracy-based benchmark by minimizing the logistic loss from Eq. (1) would exactly recover the benchmark accuracies (up to a constant shift). In contrast, the exponential used in the standard BT-model would ensure the benchmark would not exactly recover the accuracies. Thus, the ratings obtained with this model would be more directly comparable with accuracies from standard benchmarks.

**Comparison** Comparing these models is challenging because the Rao-Kupper and Davidson models include an additional draw prediction. For predictive purposes, we are only interested in the logistic loss $\mathcal{L}$ from Eq. (1) to determine whether the additional complexity of the Rao-Kupper and Davidson models reduces the value of $\mathcal{L}$ on an unknown test set. Using data from the Chatbot Arena (Chiang et al., 2024b), we compute $\mathcal{L}$ for various training set sizes. For the Davidson and Rao-Kupper models, we add $0.5P(i \simeq j|\gamma_i, \gamma_j)$ to both $P(i \succ j|\gamma_i, \gamma_j)$ and $P(j \succ i|\gamma_i, \gamma_j)$.

First of all, we see that the approximate Glicko system (Glickman, 2002) performs by far the worst, as expected. Using approximate systems for LLM evaluation is not recommended, as these systems were designed for time-varying, large-scale rating systems for multiple million players.

Results are shown in Fig. 4. The Roa-Kupper model performs the worst, while the accuracy-based model is only slightly worse than the remaining two. Finally, both the Davidson and BT-model perform almost identically. Due to the extra complexity of the Davidson model and the more frequent use of the BT-model for LLMs, we decided to use the BT-model as a default for POLYRATING.

## C  MODEL VERSIONS

Models are iteratively improved through time. Evaluating these iterations is essential for tracking performance changes and understanding the impact of updates. For developers who create multiple versions of a model simultaneously or in quick succession, it is essential to evaluate the relative performance of these versions as cost-effectively as possible. However, this process can be computationally expensive, especially when human evaluations are used. This section describes how POLYRATING enables more efficient evaluation of model versions.

### C.1  INCORPORATING MODEL VERSIONS

Model versions evolve over time, much like human capabilities in competitive games. Rating systems often reflect this evolution by introducing time-dependent factors, such as the method proposed by Coulom (2008), which uses a time-dependent Bradley-Terry (BT) model. This approach incorporates a Gaussian prior on consecutive ratings, ensuring they do not shift arbitrarily over time:

$$R_{t+1} - R_t \sim \mathcal{N}(0, \sigma^2)$$

Unlike human capabilities, model updates are irregular. Some models may never be updated, while others evolve rapidly. However, Coulom (2008) assumes that the ratings are updated at regular intervals. To remove this assumption, we modify the prior by only updating ratings upon a new release. For two versions $v_1$ and $v_2$, we impose a regularizing prior on the base rating:

$$R_{\text{base}}^{v_2} - R_{\text{base}}^{v_1} \sim \mathcal{N}(0, \sigma^2)$$

This adjustment enables faster convergence by constraining rating shifts between versions without requiring regularity. Using this prior, the loss function for evaluating versions becomes:

$$\mathcal{L}_{\text{version}}(D, \mathbf{R}_{\text{base}}, \boldsymbol{\alpha}, \boldsymbol{\beta}) = \mathcal{L}_{\text{full}}(D, \mathbf{R}_{\text{base}}, \boldsymbol{\alpha}, \boldsymbol{\beta}) + \sum_{(v_1, v_2) \in V_{\mathcal{M}}} \frac{(R_{\text{base}}^{v_1} - R_{\text{base}}^{v_2})^2}{2\sigma^2} \quad (4)$$

Here, $V_{\mathcal{M}}$ represents all pairs of consecutive model versions. Thus, if a model has versions $v_1, \ldots, v_n$, then $\{(v_1, v_2), \ldots, (v_{n-1}, v_n)\} \subset V_{\mathcal{M}}$. Note that we only consider versions of minor updates. For major updates (e.g., GPT-3.5 to GPT-4), the update is not included in $V_{\mathcal{M}}$ as the rating of the new model will likely be substantially higher and is not related to the previous model anymore. This loss function can be optimized using the same techniques as described in §3. Therefore, no additional adjustments are needed to evaluate model versions using POLYRATING.

### C.2  EVALUATING MODEL VERSIONS

We evaluate whether POLYRATING can effectively reduce the evaluation costs associated with assessing new model versions. To do this, we adopt an experimental setup similar to the one described in §4. Using the Chatbot Arena dataset, we first identify models with multiple versions to construct the set $V_{\mathcal{M}}$. Out of the 129 models in the dataset, 43 are subsequent versions of earlier models. We then split the dataset into two parts: one containing games involving these subsequent versions, and the other including all remaining games. As before, the latter set is used for both the univariate method and POLYRATING, while the former set is further divided into training and test splits.

Next, we vary the number of games available in the training set for the subsequent versions and compare how quickly the univariate method and POLYRATING converge on the test set. This approach allows us to evaluate how efficiently the methods handle

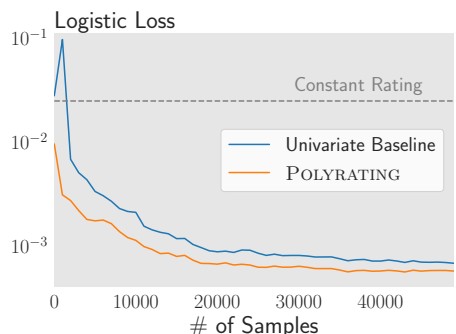

Figure 5: Convergence rate of the univariate method and POLYRATING when evaluating model versions. The x-axis represents the number of games available in the training set associated with the subsequent versions, while the y-axis represents the loss of the univariate method and POLYRATING.

new models, assuming the first version of each model has already been evaluated. The results of this experiment are shown in Fig. 5. When obtaining 10,000 samples, POLYRATING shows a 38% improvement in sample efficiency compared to the univariate method. These results indicate that POLYRATING significantly reduces the evaluation costs associated with new model versions.

## D    PROOFS

We provide the proofs for the theorems mentioned in the main text here.

We first prove the convexity of the optimization objective in Eq. (3).

**Theorem 1** (Convexity of the Optimization Objective). *The optimization objective in Eq. (3) is convex and twice differentiable.*

*Proof.* Twice differentiability follows immediately from the twice differentiability of the logistic loss and the squared penalty term. To show convexity, we make use of the following well-known facts about convex functions:

- The sum of two convex functions is convex.

- The composition of a convex function with an affine function is convex.

Since the logistic loss $f(x) = -\log(1 + \exp(x))$ is convex, and since POLYRATING relies on a linear combination of parameters in the loss function, the logistic loss is convex in these parameters. The squared penalty term is also convex, as it is a sum of squared terms with a positive quadratic coefficient. The sum of two convex functions is convex, so the optimization objective is convex.    □

Further, we show the optimality of POLYRATING by showing it converges to the same optimal rating as the univariate approach when fitted on multiple tasks at the same time.

For this purpose, suppose we have a task for which we want to obtain a separate rating. Specifically, let $D$ be a dataset of games between models. Let $D_{\neg\text{task}} \subset D$, resp. $D_{\text{task}} \subset D$, be the set of games not belonging to, resp. belonging to, the task of interest. We show that as $|D| \to \infty$, the rating obtained by individually fitting the tasks is equivalent to the rating obtained by fitting all tasks simultaneously using POLYRATING. Intuitively, the extra prior term in POLYRATING will be of less importance as the number of games in the task of interest increases, and the ratings will converge to the same optimal rating.

**Theorem 2** (Equivalence of Ratings). *Let $D$ be a set of i.i.d. games between models $m_0, \dots, m_{k-1}$. Let $D_{task} \subset D$ and $D_{\neg task} \subset D$ be as defined above. Let $\mathbf{R}_{task}$, resp. $\mathbf{R}_{\neg task}$, be the rating obtained by fitting the games in $D_{task}$, resp. $D_{\neg task}$, using the optimal univariate rating system. Let $\mathbf{R}'$ be the rating obtained by fitting all games in $D$ simultaneously using POLYRATING with the formula $R'^m(g) = R'^m_{\neg task} + \beta_1^m [\![g \in D_{task}]\!]$ and define $R'^m_{task} = R'^m_{\neg task} + \beta_1^m$. Finally, let the priors on respectively $R'^m_{\neg task}$ and $\beta_1^m$ be $\mathcal{N}(0, \sigma^2_{\neg task})$ and $\mathcal{N}(0, \sigma_1^2)$. Then, as $|D_{\neg task}| \to \infty$ and $|D_{task}| \to \infty$, $\mathbf{R}_{task}$ and $\mathbf{R}'_{task}$ will, up to a constant difference, converge to the same optimal rating $\mathbf{R}^*_{task}$ if all optimal ratings are finite. Similarly, $\mathbf{R}_{\neg task}$ and $\mathbf{R}'_{\neg task}$ will, up to a constant difference, converge to the same optimal rating $\mathbf{R}^*_{\neg task}$ if all optimal ratings are finite.*

To prove the theorem, we first need several lemmas.

**Lemma 1** (Shift-Invarance of Optimal Ratings). *Let $D$ be a set of games between models $m_0, \dots, m_{k-1}$ where there exists one model that has played all other models at least once. If $\mathbf{R}_1$ and $\mathbf{R}_2$ both minimize the logistic loss $\mathcal{L}(D, \mathbf{R})$ for $D$, then $\mathbf{R}_1 - \mathbf{R}_2$ is a constant vector.*

*Proof.* Without loss of generality, we can assume that the first model is the model that has played all other models at least once. We first note that for any constant $c \in \mathbb{R}$, it holds that $\mathcal{L}(D, \mathbf{R} + c) = \mathcal{L}(D, \mathbf{R})$. Therefore, we can assume that the first element of each vector, namely $\mathbf{R}_1^0$ and $\mathbf{R}_2^0$, are both zero by applying a constant shift to both. We now show that $\mathbf{R}_1 = \mathbf{R}_2$.

We do so by proving that the function $F(x_1, \dots x_{k-1}) = \mathcal{L}(D, (0, x_1, ..., x_{k-1}))$ is strictly convex. Since strictly convex functions have a unique minimum, this implies that $\mathbf{R}_1 = \mathbf{R}_2$. We show strict convexity by computing the Hessian and showing that it is diagonally dominant with strictly positive diagonal elements. By Gershgorin circle theorem, this implies that the Hessian cannot have eigenvalues equal to zero, and is therefore positive definite. Since the Hessian is positive definite, the function is strictly convex, and the result follows.

We compute the diagonal terms of the Hessian of $F$. We denote by $D_{\{i,j\}}$ all games where one model is $m_i$ and the other model is $m_j$. We slightly change the notation such that the game result

$g_r \in D_{\{i,j\}}$ indicates whether $m_i$ won or lost, no matter the order of the models. We define $x_0 = 0$ and drop the division by 400 for convenience. We have:

$$F(x_1, \ldots x_{k-1}) = \sum_{i=0}^{k-1} \sum_{j=0}^{k-1} \sum_{g \in D_{\{i,j\}}} g_r \log(1 + \exp(x_j - x_i))$$

Thus,

$$\frac{\partial^2 F}{\partial x_i^2} = \sum_{j=0}^{k-1} \sum_{g \in D_{\{i,j\}}} g_r \frac{\exp(x_j - x_i)}{(1 + \exp(x_j - x_i))^2} + (1 - g_r) \frac{\exp(x_i - x_j)}{(1 + \exp(x_i - x_j))^2}$$

$$= \sum_{g \in D_{\{i,0\}}} g_r \frac{\exp(-x_i)}{(1 + \exp(-x_i))^2} + (1 - g_r) \frac{\exp(x_i)}{(1 + \exp(x_i))^2} + \sum_{j=1}^{k-1} -\frac{\partial^2 F}{\partial x_i \partial x_j}$$

Since all terms in the sum are positive, the diagonal terms of the Hessian are strictly positive. Furthermore, the Hessian is diagonally dominant as the last sum is the sum over all off-diagonal terms in the same column and the first sum is strictly positive since $m_i$ has played at least one game against $m_0$. Thus, the Hessian is positive definite, and the function is strictly convex. $\qquad \square$

**Lemma 2** (Limit Exists and Is Finite). *Let $D$ be a set of i.i.d. games between models $m_0, \ldots, m_{k-1}$. Furthermore, assume all ratings are bounded. Then,*

$$\lim_{|D| \to \infty} \frac{1}{|D|} \min_{\mathbf{R}} \mathcal{L}(D, \mathbf{R}) \tag{5}$$

*almost surely uniformly converges to $\mathbb{E}_g(\mathcal{L}(g, \mathbf{R}))$. Furthermore,*

$$\lim_{|D| \to \infty} \operatorname*{arg\,min}_{\mathbf{R}, \mathbf{R}_0 = 0} \mathcal{L}(D, \mathbf{R}) \tag{6}$$

*almost surely exists and converges to the optimal rating.*

*Proof.* Let $D_n$ be the first $n$ games in $D$. We show that the functions $\mathcal{L}_n : \mathbb{R}^k \to \mathbb{R}$ defined by $\mathcal{L}_n(\mathbf{R}) = \frac{1}{n} \mathcal{L}(D_n, \mathbf{R})$ converge uniformly to $\mathbb{E}_g(\mathcal{L}(g, \mathbf{R}))$.

Thus, for any given $\epsilon > 0$ we need to prove the existence of an $N$ such that for all $n > N$ and all $\mathbf{R}$, $|\mathcal{L}_n(\mathbf{R}) - \mathbb{E}_g(\mathcal{L}(g, \mathbf{R}))| < \epsilon$. Let $\epsilon > 0$ be chosen arbitrarily. We can group games with the same models together in the notation for $\mathcal{L}_n$. More specifically, let $w_{i,j}^{(n)}$ denote the weight of the coefficient associated with $\log(1 + \exp(-R_i^m + R_j^m))$. Then we can write:

$$\frac{\mathcal{L}_n(\mathbf{R})}{n} = \sum_{i=1}^{k} \sum_{j=1}^{k} \frac{w_{i,j}^{(n)}}{n} \log(1 + \exp(-R_i^m + R_j^m))$$

Furthermore, we can write the expected value of the loss as:

$$\mathbb{E}_g(\mathcal{L}(g, \mathbf{R})) = \sum_{i=1}^{k} \sum_{j=1}^{k} (P(i \succ j) + 0.5 \cdot P(i \simeq j)) \cdot P(g \in D_{\{i,j\}}) \log(1 + \exp(-R_i^m + R_j^m))$$

$$:= \sum_{i=1}^{k} \sum_{j=1}^{k} P_{i,j} \log(1 + \exp(-R_i^m + R_j^m))$$

where $\simeq$ denotes a draw.

Thus, we obtain:

$$\frac{\mathcal{L}_n(\mathbf{R})}{n} - \mathbb{E}_g(\mathcal{L}(g, \mathbf{R})) = \sum_{i=1}^{k} \sum_{j=1}^{k} \left( \frac{w_{i,j}^{(n)}}{n} - P_{i,j} \right) \cdot \log(1 + \exp(-R_i^m + R_j^m))$$

By the strong law of large numbers, the weights $w_{i,j}^{(n)}/n$ converge almost surely to the expected value of the weights, i.e. $P_{i,j}$. Furthermore, since the ratings are finite, $\log(1 + \exp(-R_i^m + R_j^m))$ can be bounded by a constant $B$. Thus, for any $\epsilon > 0$, there exists an $N$ such that for all $n > N$, $|\frac{w_{i,j}^{(n)}}{n} - P_{i,j}| < \frac{\epsilon}{Bk^2}$ for all $i, j$. Then, almost surely,

$$
\left| \frac{\mathcal{L}_n(\mathbf{R})}{n} - \mathbb{E}_g(\mathcal{L}(g, \mathbf{R})) \right| \leqslant \sum_{i=1}^{k} \sum_{j=1}^{k} \left| \frac{w_{i,j}^{(n)}}{n} - P_{i,j} \right| B
$$

$$
\leqslant \frac{\epsilon}{Bk^2} \cdot Bk^2 = \epsilon,
$$

proving the first part of the lemma.

For the second part, we note that Lemma 1 implies that $\arg\min_{\mathbf{R}, \mathbf{R}_0 = 0} \mathcal{L}_n(\mathbf{R})$ has a unique solution. By uniform convergence on a compact domain of continuous functions $\mathcal{L}_n$, we thus have that the limit of the minimizers of $\mathcal{L}_n$ is the minimizer of the expected loss, and the result follows. $\quad\square$

Now, we can prove Theorem 2.

*Proof.* We prove that $\mathbf{R}_{\text{task}}$ and $\mathbf{R}'_{\text{task}}$ will converge to the same optimal rating $\mathbf{R}^*_{\text{task}}$ assuming that $\mathbf{R}_{\text{task},0} = \mathbf{R}'_{\text{task},0} = \mathbf{R}^*_{\text{task},0} = 0$ which can be assumed due to shift-invariance. The other implication is proven equivalently.

Let $D_{\text{task}}^{(n)}$ denote the first $n$ elements of $D_{\text{task}}$ and $\mathbf{R}_{\text{task}}^{(n)}$ the optimal solutions found when fitting using $D_{\text{task}}^{(n)}$. Note that we leave the size of $D_{\neg\text{task}}$ in the sequence unspecified since it does not matter for this part of the proof. We note that POLYRATING optimizes the loss

$$
\mathcal{L}_{\text{full}}^{(n)}(D^{(n)}, \mathbf{R}') = \frac{1}{n}\mathcal{L}(D_{\neg\text{task}}, \mathbf{R}'_{\neg\text{task}}) + \frac{1}{n}\mathcal{L}(D_{\text{task}}^{(n)}, \mathbf{R}'_{\text{task}}) + \frac{1}{n}\sum_{j=0}^{d} \frac{\mathbf{R}'^2_{\neg\text{task},j}}{2\sigma_{\neg\text{task}}^2} + \frac{1}{n}\sum_{j=0}^{d} \frac{\beta_{1,j}^2}{2\sigma_1^2} \quad (7)
$$

with optimal solution $\mathbf{R}'^{(n)}$. Suppose now that $\mathbf{R}'^{(n)}_{\text{task}}$ does not converge to $\mathbf{R}^*_{\text{task}}$. Then there exists a subsequence $n_i$ and a $\delta > 0$ such that $||\mathbf{R}'^{(n_i)}_{\text{task}} - \mathbf{R}^*_{\text{task}}|| > \delta$. Without loss of generalization, we can assume this subsequence is the full sequence.

By Lemma 2, we know that $\frac{1}{n}\mathcal{L}(D_{\text{task}}^{(n)}, \mathbf{R})$ uniformly converges to the function $\mathcal{L}^*(\mathbf{R}) := \mathbb{E}_{g_{\text{task}}}(\mathcal{L}(g_{\text{task}}, \mathbf{R}))$ which has $\mathbf{R}^*_{\text{task}}$ as minimizer. By Lemma 1, $\mathcal{L}^*$ is continuous and has a unique minimum that satisfies $\mathbf{R}^*_{\text{task},0} = 0$. Therefore, there exists an $\epsilon > 0$ such that for all ratings $\mathbf{R}$ with $\mathbf{R}_0 = 0$ the following is true:

$$
||\mathbf{R} - \mathbf{R}^*_{\text{task}}|| > \delta \Rightarrow \mathcal{L}^*(\mathbf{R}) - \mathcal{L}^*(\mathbf{R}^*_{\text{task}}) > \epsilon. \quad (8)
$$

Since $\mathbf{R}_{\text{task}}^{(n)}$ converges to $\mathbf{R}^*_{\text{task}}$ by Lemma 2, we know there is an $n_0 > 0$ such that for each $n > n_0$,

$$
\mathcal{L}^*(\mathbf{R}_{\text{task}}^{(n)}) - \mathcal{L}^*(\mathbf{R}^*_{\text{task}}) < \frac{\epsilon}{4}. \quad (9)
$$

Furthermore, due to the uniform convergence of the loss, there exists an $n_1 > 0$ such that for all $n > n_1$ and all $\mathbf{R}$,

$$
\left| \frac{1}{n}\mathcal{L}(D_{\text{task}}^{(n)}, \mathbf{R}) - \mathcal{L}^*(\mathbf{R}) \right| < \frac{\epsilon}{4}. \quad (10)
$$

Finally, there exists an $n_2 > 0$ such that for all $n > n_2$,

$$
\left| \frac{d+1}{n}\frac{B^2}{2\sigma_{\neg\text{task}}^2} + \frac{d+1}{n}\frac{4B^2}{2\sigma_1^2} \right| < \frac{\epsilon}{4} \quad (11)
$$

where $B$ is the upper bound for all ratings.

However, we can now define $\mathbf{R}''^{(n)}_{\neg\text{task}} = \mathbf{R}'^{(n)}_{\neg\text{task}}$ and $\beta''^{(n)} = \mathbf{R}_{\text{task}}^{(n)} - \mathbf{R}'^{(n)}_{\neg\text{task}}$. The inequalities above imply that for all $n > \max(n_0, n_1, n_2)$, $\mathcal{L}_{\text{full}}^{(n)}(D^{(n)}, \mathbf{R}''^{(n)}) < \mathcal{L}_{\text{full}}^{(n)}(D^{(n)}, \mathbf{R}'^{(n)})$, since

$$\frac{1}{n}\mathcal{L}(D_{\text{task}}^{(n)}, \mathbf{R'}_{\text{task}}) - \frac{1}{n}\mathcal{L}(D_{\text{task}}^{(n)}, \mathbf{R}_{\text{task}}) > \mathcal{L}^*(\mathbf{R'}_{\text{task}}) - \frac{\epsilon}{4} - \mathcal{L}^*(\mathbf{R}_{\text{task}}) - \frac{\epsilon}{4}$$
$$= \mathcal{L}^*(\mathbf{R'}_{\text{task}}) - \mathcal{L}^*(\mathbf{R}_{\text{task}}^*) + \mathcal{L}^*(\mathbf{R}_{\text{task}}^*) - \mathcal{L}^*(\mathbf{R}_{\text{task}}) - \frac{\epsilon}{2}$$
$$> \epsilon - \frac{\epsilon}{4} - \frac{\epsilon}{2} = \frac{\epsilon}{4}$$

where the first inequality follows from Eq. (10) and the last from Eq. (8) and Eq. (9). Since Eq. (11) ensures that the difference in the bias term can at most differ by $\epsilon/4$, we find $\mathcal{L}_{\text{full}}^{(n)}(D^{(n)}, \mathbf{R''}^{(n)}) < \mathcal{L}_{\text{full}}^{(n)}(D^{(n)}, \mathbf{R'}^{(n)})$. Therefore, $\mathbf{R'}_{\text{task}}^{(n)}$ cannot be the optimal solution, which is a contradiction to the optimality of $\mathbf{R'}^{(n)}$. Therefore, $\mathbf{R'}_{\text{task}}$ must converge to $\mathbf{R}_{\text{task}}^*$.

$\square$

# E  EXPERIMENTAL DETAILS

In this section, we provide detailed descriptions of the biases and tasks used in our experiments. In Table 4 we describe the biases used in §4.1 in more detail. In Table 5 we describe the tasks used in §4 in more detail. Finally, we note that any run using POLYRATING took at most 6 hours on a single CPU, even for huge datasets with a million samples, 100 models and 10 tasks.

We also briefly explain how we adjust the win rates of traditional benchmarks to improve sample efficiency of human evaluation, as discussed in §4.2. As detailed in the accuracy-based model in App. B, the win rate of $m_1$ over $m_2$ in a traditional benchmark can be written as

$$P(i \succ j | m_i, m_j) = \frac{1}{2}\Big(1 + \text{Acc}_D(m_1) - \text{Acc}_D(m_2)\Big)$$
$$P(j \succ i | m_i, m_j) = \frac{1}{2}\Big(1 + \text{Acc}_D(m_2) - \text{Acc}_D(m_1)\Big).$$

where $\text{Acc}_D$ is the accuracy function.

Benchmarks often exhibit significant variation in accuracy differences between models. For instance, in some benchmarks, models may have closely aligned accuracies, while in others, the differences may be substantial. This variation affects the win rate estimates between models. To address this, we introduce a parameter $\mathcal{W}$, which allows us to adjust the scale of win rates. The adjusted win rates are modeled as:

$$P(i \succ j | m_i, m_j) = \min\Big(1, \frac{\mathcal{W}}{2}\Big(1 + \text{Acc}_D(m_1) - \text{Acc}_D(m_2)\Big)\Big)$$
$$P(j \succ i | m_i, m_j) = 1 - P(i \succ j | m_i, m_j).$$

This adjustment ensures that we can control the scale of win rates, mitigating the issue of varying accuracy differences. We optimize the hyperparameter $\mathcal{W}$ using human evaluations from the training data. Specifically, we fit a univariate rating model using win rates for a given $\mathcal{W}$ on the classical benchmark and evaluate the logistic loss of the resulting ratings on the training data. The parameter with the lowest logistic loss is selected. Importantly, we do not use any unknown test data during this optimization process, ensuring that our approach can be applied in practical scenarios without compromising the integrity of the evaluation.

---

[1]https://huggingface.co/s-nlp/roberta-base-formality-ranker
[2]https://huggingface.co/cardiffnlp/twitter-roberta-base-sentiment-latest

Table 4: Overview of all biases used in §4.1. The table contains a description of the bias and a functional form of the bias. Scaling constant were introduced in these functional forms to ensure that all biases output values within the same order of magnitude.

| Bias | Description | Functional Form |
|---|---|---|
| Length | Measures the length of a model answer for a given question. | $f(g, i) = \log_{10}(\text{length}(g_{m_i}(g_p)))$ |
| Position | Computes the order of the model in the game. | $f(g, i) = [\![i = 1]\!]$ |
| Formality | Computes the formality of an answer computed by a popular formality classifier (Babakov et al., 2023).[1] | $f(g, i) = \mathcal{M}(g_{m_i}(g_p))_1$ |
| Sentiment | Computes the sentiment of an answer computed by a popular sentiment classifier (Camacho-collados et al., 2022).[2] | $f(g, i) = \mathcal{M}(g_{m_i}(g_p))_2$ |
| Repetitiveness | Computes the repetitiveness of the answer by computing the percentage of non-unique words in the answer. | $f(g, i) = 5 \cdot \frac{\text{\# of repeated words in } g_{m_i}(g_p)}{\text{\# of words in } g_{m_i}(g_p)}$ |
| Readability | Computes the Flesch Reading Ease score (Kincaid et al., 1975) of an answer. | $f(g, i) = \min(1, \max(0, \frac{\text{Flesch}(g_{m_i}(g_p))}{100}))$ |

Table 5: Overview of all tasks used in §4. For each task, we use the same data as the actual Chatbot Arena (Chiang et al., 2024b)

| Task | Description |
|---|---|
| English | Questions that are in English. |
| Chinese | Questions that are in Chinese. |
| Hardness | Questions that are considered hard by the Chatbot Arena. These are questions that are classified as being in at least six of the following seven categories: specific, requires domain knowledge, is complex, requires problem-solving, requires creative thinking, requires technical accuracy, is a real-world question. |
| Code | Questions that require code to be answered. |
| LLM | Whether the judge is a language model. |

Table 6: Fitted coefficients for the biases and their average influence on the ratings of the models for both human and LLM-based evaluation. The functional form of $f_{\text{bias}}$ used for each bias can be found in App. E. The influence is computed as $\mathbb{E}_g(\alpha_{\text{bias}} \cdot |f_{\text{bias}}(g, 0) - f_{\text{bias}}(g, 1)|)$ and indicates the average influence the bias has on the rating of models for specific games. Errors shown are $95\%$ pivot intervals computed using bootstrapping.

(a) Human Evaluation

| Bias | Coefficient $(\alpha)$ | Influence $(\mathbb{E})$ |
|---|---|---|
| Length | $130.74^{+7.9}_{-7.3}$ | $40.84^{+2.5}_{-2.3}$ |
| Position | $2.70^{+2.3}_{-2.4}$ | $2.70^{+2.3}_{-2.4}$ |
| Formality | $-119.89^{+11.6}_{-11.4}$ | $-15.17^{+1.5}_{-1.4}$ |
| Sentiment | $57.42^{+10.1}_{-10.9}$ | $7.90^{+1.4}_{-1.5}$ |
| Repetitiveness | $-22.10^{+8.5}_{-8.4}$ | $-4.64^{+1.8}_{-1.8}$ |
| Readability | $72.93^{+11.0}_{-11.6}$ | $10.75^{+1.6}_{-1.7}$ |

(b) LLM-based Evaluation

| Bias | Coefficient $(\alpha)$ | Influence $(\mathbb{E})$ |
|---|---|---|
| Length | $251.87^{+7.3}_{-6.8}$ | $48.48^{+1.4}_{-1.3}$ |
| Position | $37.53^{+1.1}_{-1.2}$ | $37.53^{+1.1}_{-1.2}$ |
| Formality | $-37.56^{+6.7}_{-7.2}$ | $-4.31^{+0.8}_{-0.8}$ |
| Sentiment | $4.31^{+6.1}_{-6.7}$ | $0.43^{+0.6}_{-0.7}$ |
| Repetitiveness | $75.04^{+8.8}_{-7.3}$ | $9.12^{+1.1}_{-0.9}$ |
| Readability | $-32.56^{+8.1}_{-7.9}$ | $-3.92^{+1.0}_{-0.9}$ |

# F    DETAILED RESULTS

In Table 6, we show Table 1 with the adjusted confidence intervals computed using pivot intervals instead of $2\sigma$ intervals.

The full multidimensional leaderboard fitted using POLYRATING on Chatbot Arena data (Chiang et al., 2024b) can be found in Table 7. The full leaderboard fitted using a unidimensional approach can be found in Table 8.

Table 7: Leaderboard of human evaluation with modifiers fitted with POLYRATING. Indicated deviations are 95% pivot intervals determined using bootstrapping.

| Rank | Model Name | Rating | English | Chinese | Hardness | Code |
|---|---|---|---|---|---|---|
| 1 | gpt-4o-2024-05-13 | $1297^{+4.5}_{-4.2}$ | $-13^{+4.3}_{-10.3}$ | $-3^{+8.8}_{-8.2}$ | $13^{+10.0}_{-4.8}$ | $15^{+10.4}_{-5.7}$ |
| 2 | claude-3-5-sonnet-20240620 | $1286^{+6.7}_{-7.4}$ | $-29^{+8.2}_{-13.9}$ | $-19^{+15.0}_{-12.9}$ | $14^{+13.6}_{-8.5}$ | $44^{+14.9}_{-10.6}$ |
| 3 | gemini-advanced-0514 | $1285^{+4.5}_{-5.1}$ | $-26^{+4.5}_{-10.7}$ | $8^{+9.2}_{-9.4}$ | $3^{+10.3}_{-5.6}$ | $2^{+11.2}_{-6.9}$ |
| 4 | gemini-1.5-pro-api-0514 | $1273^{+4.8}_{-4.5}$ | $-20^{+4.8}_{-10.5}$ | $19^{+10.0}_{-4.8}$ | $15^{+10.0}_{-4.8}$ | $11^{+11.3}_{-5.8}$ |
| 5 | claude-3-opus-20240229 | $1273^{+3.0}_{-2.7}$ | $-39^{+2.5}_{-9.2}$ | $1^{+6.0}_{-5.1}$ | $15^{+8.4}_{-2.9}$ | $12^{+8.4}_{-3.7}$ |
| 6 | bard-jan-24-gemini-pro | $1271^{+12.4}_{-12.6}$ | $-48^{+11.7}_{-16.9}$ | $-25^{+25.6}_{-25.8}$ | $-45^{+16.9}_{-11.7}$ | $-11^{+19.0}_{-13.4}$ |
| 7 | gpt-4-1106-preview | $1265^{+3.7}_{-4.0}$ | $-11^{+3.5}_{-9.7}$ | $-3^{+8.0}_{-7.2}$ | $9^{+8.7}_{-3.5}$ | $11^{+9.3}_{-4.8}$ |
| 8 | gemini-1.5-pro-api-0409-preview | $1264^{+4.6}_{-4.4}$ | $-2^{+4.1}_{-10.1}$ | $5^{+7.9}_{-8.5}$ | $-4^{+9.8}_{-4.8}$ | $-12^{+10.8}_{-5.5}$ |
| 9 | gpt-4-turbo-2024-04-09 | $1258^{+3.8}_{-3.4}$ | $3^{+3.6}_{-9.8}$ | $4^{+7.4}_{-6.6}$ | $11^{+8.9}_{-3.8}$ | $15^{+9.6}_{-4.9}$ |
| 10 | gpt-4-0125-preview | $1255^{+3.6}_{-3.6}$ | $-5^{+3.5}_{-9.6}$ | $1^{+7.3}_{-6.1}$ | $14^{+9.0}_{-3.9}$ | $2^{+9.5}_{-4.7}$ |
| 11 | gemini-1.5-flash-api-0514 | $1243^{+4.7}_{-4.5}$ | $-21^{+4.5}_{-10.5}$ | $8^{+9.5}_{-9.7}$ | $9^{+10.4}_{-5.5}$ | $15^{+10.8}_{-6.7}$ |
| 12 | yi-large-preview | $1237^{+4.4}_{-4.4}$ | $-4^{+4.1}_{-10.6}$ | $37^{+8.8}_{-8.6}$ | $17^{+10.5}_{-4.8}$ | $11^{+11.0}_{-5.8}$ |
| 13 | gemma-2-27b-it | $1232^{+11.7}_{-11.4}$ | $-10^{+12.2}_{-17.6}$ | $-1^{+23.2}_{-19.6}$ | $-15^{+18.0}_{-13.6}$ | $7^{+19.9}_{-15.7}$ |
| 14 | yi-large | $1222^{+8.8}_{-7.7}$ | $-3^{+8.5}_{-15.5}$ | $9^{+17.9}_{-16.5}$ | $11^{+15.4}_{-10.4}$ | $21^{+18.2}_{-12.5}$ |
| 15 | nemotron-4-340b-instruct | $1222^{+6.5}_{-7.2}$ | $-13^{+7.6}_{-13.1}$ | $7^{+13.9}_{-12.4}$ | $9^{+13.2}_{-8.5}$ | $-6^{+14.7}_{-9.4}$ |
| 16 | claude-3-sonnet-20240229 | $1220^{+3.3}_{-3.2}$ | $-26^{+2.8}_{-9.4}$ | $-15^{+8.3}_{-5.5}$ | $6^{+8.3}_{-3.6}$ | $26^{+9.2}_{-3.7}$ |
| 17 | command-r-plus | $1214^{+3.9}_{-3.5}$ | $-18^{+3.4}_{-9.4}$ | $6^{+6.7}_{-6.4}$ | $-7^{+9.0}_{-3.7}$ | $-14^{+9.3}_{-4.4}$ |
| 18 | gpt-4-0314 | $1213^{+4.5}_{-4.5}$ | $-29^{+4.3}_{-10.4}$ | $-13^{+8.7}_{-7.9}$ | $23^{+9.7}_{-4.6}$ | $9^{+10.5}_{-4.9}$ |
| 19 | reka-core-20240501 | $1212^{+3.6}_{-3.9}$ | $-11^{+3.7}_{-9.8}$ | $9^{+8.1}_{-7.6}$ | $6^{+9.3}_{-4.7}$ | $-2^{+10.4}_{-4.6}$ |
| 20 | claude-3-haiku-20240307 | $1209^{+3.5}_{-3.4}$ | $-30^{+3.2}_{-9.2}$ | $-36^{+6.0}_{-5.2}$ | $10^{+8.4}_{-3.4}$ | $16^{+9.0}_{-3.6}$ |
| 21 | gemma-2-9b-it | $1204^{+11.0}_{-10.7}$ | $-17^{+12.6}_{-17.5}$ | $-1^{+21.4}_{-22.1}$ | $3^{+18.6}_{-12.6}$ | $-14^{+20.3}_{-16.3}$ |
| 22 | glm-4-0520 | $1202^{+10.0}_{-9.6}$ | $8^{+10.8}_{-15.8}$ | $49^{+17.8}_{-17.7}$ | $19^{+15.7}_{-11.1}$ | $12^{+17.4}_{-12.5}$ |
| 23 | gpt-4-0613 | $1191^{+3.8}_{-3.7}$ | $-22^{+3.6}_{-9.5}$ | $-41^{+7.6}_{-6.5}$ | $18^{+8.5}_{-3.7}$ | $1^{+9.5}_{-4.2}$ |
| 24 | claude-1 | $1190^{+8.3}_{-8.2}$ | $-30^{+7.9}_{-14.5}$ | $-18^{+16.9}_{-17.0}$ | $-19^{+13.3}_{-7.6}$ | $4^{+14.8}_{-9.4}$ |
| 25 | reka-flash-preview-20240611 | $1188^{+7.7}_{-7.6}$ | $-15^{+7.3}_{-13.8}$ | $-5^{+14.1}_{-15.0}$ | $-10^{+13.8}_{-9.0}$ | $7^{+15.4}_{-10.6}$ |
| 26 | llama-3-70b-instruct | $1187^{+3.0}_{-2.7}$ | $67^{+2.3}_{-9.0}$ | $-51^{+5.7}_{-5.0}$ | $-1^{+7.8}_{-2.5}$ | $-10^{+8.6}_{-3.3}$ |
| 27 | qwen-max-0428 | $1187^{+5.4}_{-5.1}$ | $3^{+5.4}_{-11.6}$ | $62^{+12.9}_{-11.3}$ | $11^{+11.2}_{-7.1}$ | $8^{+12.3}_{-7.9}$ |
| 28 | qwen2-72b-instruct | $1182^{+6.0}_{-6.0}$ | $9^{+5.4}_{-12.9}$ | $68^{+11.9}_{-11.0}$ | $13^{+11.6}_{-6.8}$ | $-3^{+13.4}_{-7.6}$ |
| 29 | gemini-pro-dev-api | $1182^{+8.1}_{-7.2}$ | $-38^{+7.2}_{-13.6}$ | $-23^{+13.9}_{-14.0}$ | $-11^{+13.3}_{-8.7}$ | $-23^{+14.3}_{-9.2}$ |
| 30 | deepseek-coder-v2 | $1181^{+9.4}_{-9.3}$ | $-34^{+9.6}_{-15.6}$ | $16^{+17.3}_{-17.5}$ | $43^{+15.5}_{-11.0}$ | $64^{+18.1}_{-11.5}$ |
| 31 | reka-flash-21b-20240226-online | $1176^{+6.9}_{-7.1}$ | $-12^{+6.9}_{-13.3}$ | $-10^{+13.3}_{-12.1}$ | $-2^{+13.0}_{-8.1}$ | $1^{+13.4}_{-9.4}$ |
| 32 | command-r | $1175^{+4.2}_{-3.9}$ | $-15^{+3.9}_{-9.9}$ | $12^{+7.5}_{-6.8}$ | $-23^{+9.3}_{-4.6}$ | $-9^{+10.6}_{-5.3}$ |
| 33 | reka-flash-21b-20240226 | $1170^{+5.6}_{-5.2}$ | $-16^{+5.2}_{-11.7}$ | $-12^{+10.1}_{-11.7}$ | $-6^{+11.7}_{-6.4}$ | $5^{+11.5}_{-7.2}$ |
| 34 | claude-2.0 | $1164^{+10.8}_{-10.4}$ | $-26^{+9.9}_{-15.8}$ | $-8^{+24.1}_{-23.1}$ | $3^{+15.1}_{-10.9}$ | $12^{+17.6}_{-13.0}$ |
| 35 | mistral-large-2402 | $1163^{+4.0}_{-4.2}$ | $2^{+3.8}_{-10.0}$ | $-32^{+7.7}_{-7.1}$ | $20^{+9.0}_{-4.2}$ | $10^{+9.9}_{-5.3}$ |
| 36 | gpt-3.5-turbo-0314 | $1162^{+19.5}_{-21.0}$ | $-56^{+19.3}_{-25.0}$ | $9^{+31.9}_{-32.4}$ | $22^{+22.7}_{-18.2}$ | $14^{+27.5}_{-21.9}$ |
| 37 | qwen1.5-110b-chat | $1161^{+4.9}_{-5.0}$ | $12^{+4.9}_{-11.0}$ | $58^{+11.1}_{-10.5}$ | $10^{+10.8}_{-7.2}$ | $11^{+12.5}_{-7.8}$ |
| 38 | gpt-3.5-turbo-0613 | $1161^{+6.2}_{-6.3}$ | $-41^{+4.9}_{-11.3}$ | $-37^{+16.2}_{-14.5}$ | $13^{+10.5}_{-6.0}$ | $21^{+12.1}_{-7.9}$ |
| 39 | claude-2.1 | $1156^{+6.3}_{-5.7}$ | $-40^{+5.3}_{-11.6}$ | $-45^{+12.2}_{-11.8}$ | $10^{+10.8}_{-5.3}$ | $22^{+11.8}_{-6.9}$ |
| 40 | mistral-next | $1153^{+11.0}_{-10.6}$ | $-20^{+9.7}_{-15.2}$ | $-50^{+20.2}_{-19.4}$ | $14^{+14.9}_{-10.4}$ | $8^{+17.7}_{-13.2}$ |
| 41 | mistral-medium | $1153^{+6.1}_{-6.0}$ | $9^{+5.2}_{-11.4}$ | $-24^{+11.1}_{-10.7}$ | $5^{+11.4}_{-5.8}$ | $11^{+13.2}_{-7.2}$ |

Table 7: (continued)

| Rank | Model Name | Rating | English | Chinese | Hardness | Code |
|---|---|---|---|---|---|---|
| 42 | mixtral-8x22b-instruct-v0.1 | $1152^{+4.9}_{-4.5}$ | $5^{+4.3}_{-10.3}$ | $-7^{+8.6}_{-9.0}$ | $12^{+10.3}_{-4.6}$ | $8^{+10.9}_{-5.9}$ |
| 43 | llama-3-8b-instruct | $1150^{+3.5}_{-3.4}$ | $48^{+3.3}_{-9.3}$ | $-41^{+6.9}_{-6.0}$ | $-16^{+8.5}_{-3.4}$ | $-4^{+8.7}_{-4.1}$ |
| 44 | glm-4-0116 | $1149^{+9.2}_{-10.7}$ | $45^{+10.5}_{-15.8}$ | $76^{+19.5}_{-18.7}$ | $21^{+17.4}_{-11.6}$ | $12^{+18.5}_{-14.0}$ |
| 45 | qwen1.5-72b-chat | $1148^{+5.4}_{-4.6}$ | $11^{+4.7}_{-10.6}$ | $58^{+9.0}_{-9.2}$ | $-1^{+10.2}_{-4.7}$ | $19^{+11.5}_{-6.3}$ |
| 46 | gpt-3.5-turbo-0125 | $1147^{+4.1}_{-3.8}$ | $-38^{+3.5}_{-10.1}$ | $-46^{+7.3}_{-7.1}$ | $11^{+9.2}_{-4.4}$ | $19^{+10.2}_{-4.5}$ |
| 47 | zephyr-orpo-141b-A35b-v0.1 | $1143^{+11.7}_{-12.2}$ | $4^{+12.2}_{-19.4}$ | $-27^{+21.3}_{-22.2}$ | $-2^{+21.5}_{-16.2}$ | $-1^{+22.5}_{-18.9}$ |
| 48 | gemini-pro | $1139^{+14.4}_{-14.6}$ | $-7^{+14.6}_{-20.4}$ | $2^{+29.9}_{-29.5}$ | $-23^{+18.7}_{-14.0}$ | $-7^{+22.2}_{-17.3}$ |
| 49 | claude-instant-1 | $1134^{+8.1}_{-8.7}$ | $-13^{+8.1}_{-14.0}$ | $-15^{+18.8}_{-19.0}$ | $7^{+13.3}_{-7.8}$ | $3^{+14.7}_{-10.7}$ |
| 50 | wizardlm-70b | $1129^{+12.7}_{-13.0}$ | $5^{+12.8}_{-17.3}$ | $-30^{+30.0}_{-28.0}$ | $-19^{+17.7}_{-13.6}$ | $-26^{+20.1}_{-16.3}$ |
| 51 | snowflake-arctic-instruct | $1126^{+5.3}_{-5.0}$ | $-13^{+5.2}_{-11.1}$ | $-3^{+9.8}_{-9.4}$ | $-13^{+10.7}_{-5.7}$ | $-11^{+12.1}_{-6.7}$ |
| 52 | qwen1.5-32b-chat | $1126^{+5.8}_{-6.0}$ | $5^{+5.8}_{-11.7}$ | $69^{+11.1}_{-10.4}$ | $8^{+11.7}_{-7.1}$ | $22^{+13.3}_{-8.9}$ |
| 53 | yi-1.5-34b-chat | $1126^{+6.4}_{-5.9}$ | $63^{+6.9}_{-12.2}$ | $103^{+12.1}_{-11.4}$ | $5^{+12.6}_{-7.6}$ | $1^{+13.5}_{-8.9}$ |
| 54 | phi-3-medium-4k-instruct | $1126^{+7.3}_{-7.4}$ | $12^{+7.9}_{-13.8}$ | $-7^{+13.0}_{-13.0}$ | $21^{+13.5}_{-8.6}$ | $5^{+15.2}_{-11.1}$ |
| 55 | tulu-2-dpo-70b | $1122^{+13.4}_{-13.7}$ | $-4^{+12.4}_{-19.0}$ | $-72^{+29.7}_{-28.7}$ | $8^{+18.6}_{-12.9}$ | $-6^{+21.1}_{-16.3}$ |
| 56 | mixtral-8x7b-instruct-v0.1 | $1114^{+4.1}_{-3.8}$ | $25^{+3.4}_{-10.2}$ | $-37^{+7.4}_{-6.9}$ | $10^{+9.3}_{-4.0}$ | $-4^{+10.2}_{-4.8}$ |
| 57 | openchat-3.5-0106 | $1114^{+8.7}_{-8.4}$ | $-3^{+8.5}_{-14.5}$ | $-3^{+17.0}_{-14.4}$ | $-11^{+14.0}_{-9.4}$ | $17^{+15.9}_{-11.5}$ |
| 58 | qwen1.5-14b-chat | $1112^{+6.4}_{-6.5}$ | $10^{+6.6}_{-12.5}$ | $57^{+11.2}_{-10.3}$ | $8^{+12.4}_{-8.1}$ | $11^{+14.2}_{-9.0}$ |
| 59 | llama2-70b-steerlm-chat | $1111^{+20.6}_{-20.2}$ | $-3^{+19.4}_{-25.7}$ | $-28^{+34.6}_{-33.8}$ | $-13^{+25.0}_{-19.7}$ | $-52^{+25.8}_{-21.6}$ |
| 60 | starling-lm-7b-beta | $1111^{+7.7}_{-7.1}$ | $19^{+7.6}_{-14.1}$ | $35^{+12.7}_{-11.3}$ | $1^{+13.0}_{-7.7}$ | $18^{+13.7}_{-10.5}$ |
| 61 | llama-2-70b-chat | $1108^{+5.8}_{-4.9}$ | $24^{+4.7}_{-10.3}$ | $-78^{+10.8}_{-10.4}$ | $-18^{+10.3}_{-5.8}$ | $-15^{+11.1}_{-7.0}$ |
| 62 | gpt-3.5-turbo-1106 | $1106^{+8.7}_{-8.8}$ | $-36^{+8.6}_{-14.7}$ | $-62^{+22.6}_{-21.8}$ | $33^{+14.7}_{-8.4}$ | $20^{+15.3}_{-11.5}$ |
| 63 | vicuna-33b | $1105^{+8.2}_{-7.9}$ | $17^{+6.6}_{-12.9}$ | $-27^{+16.3}_{-14.4}$ | $-18^{+12.6}_{-7.8}$ | $-18^{+14.1}_{-8.9}$ |
| 64 | phi-3-small-8k-instruct | $1103^{+7.0}_{-6.7}$ | $27^{+7.6}_{-12.6}$ | $-16^{+12.9}_{-12.0}$ | $13^{+13.5}_{-8.0}$ | $-5^{+15.1}_{-9.4}$ |
| 65 | openchat-3.5 | $1101^{+13.2}_{-12.9}$ | $-5^{+12.1}_{-18.3}$ | $2^{+26.4}_{-28.3}$ | $2^{+17.2}_{-11.6}$ | $-23^{+20.5}_{-16.2}$ |
| 66 | dbrx-instruct-preview | $1101^{+5.3}_{-5.3}$ | $25^{+4.9}_{-10.8}$ | $-4^{+9.6}_{-9.4}$ | $5^{+10.5}_{-5.4}$ | $17^{+12.4}_{-6.8}$ |
| 67 | yi-34b-chat | $1101^{+8.1}_{-7.9}$ | $36^{+7.5}_{-14.4}$ | $94^{+16.5}_{-16.5}$ | $-6^{+14.0}_{-9.8}$ | $-6^{+14.5}_{-10.0}$ |
| 68 | starling-lm-7b-alpha | $1099^{+11.0}_{-10.4}$ | $17^{+10.5}_{-15.8}$ | $-17^{+20.0}_{-19.9}$ | $-12^{+16.3}_{-11.9}$ | $-3^{+18.1}_{-12.6}$ |
| 69 | gemma-1.1-7b-it | $1097^{+6.0}_{-6.0}$ | $14^{+5.9}_{-11.4}$ | $-2^{+10.4}_{-10.2}$ | $-9^{+12.2}_{-6.5}$ | $3^{+12.7}_{-8.5}$ |
| 70 | pplx-70b-online | $1095^{+12.8}_{-15.4}$ | $11^{+15.2}_{-19.4}$ | $13^{+31.4}_{-29.1}$ | $-39^{+20.3}_{-13.8}$ | $-32^{+19.9}_{-16.0}$ |
| 71 | deepseek-llm-67b-chat | $1092^{+15.8}_{-17.2}$ | $0^{+16.4}_{-21.6}$ | $46^{+34.4}_{-30.4}$ | $-10^{+23.9}_{-16.0}$ | $10^{+22.9}_{-20.2}$ |
| 72 | nous-hermes-2-mixtral-8x7b-dpo | $1090^{+20.4}_{-18.6}$ | $25^{+18.3}_{-24.8}$ | $-26^{+29.7}_{-37.1}$ | $-42^{+21.5}_{-16.5}$ | $17^{+23.0}_{-21.5}$ |
| 73 | qwen1.5-7b-chat | $1086^{+13.4}_{-13.7}$ | $-4^{+13.7}_{-19.7}$ | $72^{+26.1}_{-22.7}$ | $-8^{+19.6}_{-15.3}$ | $24^{+22.5}_{-18.0}$ |
| 74 | wizardlm-13b | $1083^{+14.8}_{-14.1}$ | $5^{+12.4}_{-19.6}$ | $-10^{+26.1}_{-27.5}$ | $-41^{+19.6}_{-15.9}$ | $-13^{+23.0}_{-19.1}$ |
| 75 | llama-2-13b-chat | $1081^{+7.3}_{-7.9}$ | $12^{+7.3}_{-12.7}$ | $-58^{+17.3}_{-15.4}$ | $-10^{+12.9}_{-7.7}$ | $-9^{+15.0}_{-10.4}$ |
| 76 | qwen-14b-chat | $1081^{+15.3}_{-16.1}$ | $-35^{+16.2}_{-20.3}$ | $11^{+35.3}_{-31.7}$ | $-17^{+22.0}_{-17.9}$ | $33^{+25.2}_{-20.4}$ |
| 77 | vicuna-13b | $1077^{+8.7}_{-8.9}$ | $-15^{+8.2}_{-12.9}$ | $7^{+16.5}_{-16.9}$ | $-15^{+13.8}_{-9.1}$ | $1^{+15.2}_{-10.7}$ |
| 78 | openhermes-2.5-mistral-7b | $1075^{+15.1}_{-14.8}$ | $28^{+14.7}_{-20.9}$ | $-13^{+31.0}_{-29.5}$ | $3^{+19.7}_{-16.8}$ | $-17^{+23.8}_{-18.4}$ |
| 79 | phi-3-mini-128k-instruct | $1072^{+6.2}_{-6.3}$ | $0^{+6.4}_{-11.9}$ | $-2^{+12.1}_{-11.6}$ | $0^{+12.7}_{-6.9}$ | $-23^{+13.4}_{-8.7}$ |
| 80 | codellama-34b-instruct | $1070^{+12.0}_{-13.1}$ | $-5^{+12.3}_{-19.1}$ | $-57^{+29.8}_{-30.6}$ | $-13^{+19.3}_{-13.2}$ | $5^{+22.4}_{-17.3}$ |
| 81 | phi-3-mini-4k-instruct | $1068^{+6.2}_{-5.7}$ | $28^{+6.2}_{-12.5}$ | $-24^{+13.5}_{-11.6}$ | $15^{+12.2}_{-7.5}$ | $10^{+13.5}_{-8.0}$ |
| 82 | solar-10.7b-instruct-v1.0 | $1064^{+18.2}_{-17.8}$ | $27^{+17.9}_{-22.0}$ | $-23^{+33.4}_{-33.3}$ | $2^{+22.4}_{-17.3}$ | $-15^{+27.7}_{-21.3}$ |
| 83 | dolphin-2.2.1-mistral-7b | $1060^{+25.1}_{-24.5}$ | $30^{+24.6}_{-29.4}$ | $15^{+37.1}_{-37.3}$ | $0^{+29.8}_{-23.2}$ | $-31^{+33.5}_{-28.0}$ |
| 84 | vicuna-7b | $1058^{+16.1}_{-16.7}$ | $-36^{+15.7}_{-21.5}$ | $-37^{+28.8}_{-27.6}$ | $-3^{+19.3}_{-14.6}$ | $-18^{+21.4}_{-18.8}$ |
| 85 | falcon-180b-chat | $1056^{+29.7}_{-29.4}$ | $3^{+26.8}_{-34.2}$ | $-22^{+35.1}_{-34.5}$ | $-25^{+32.1}_{-28.4}$ | $-4^{+40.6}_{-34.0}$ |
| 86 | mistral-7b-instruct-v0.2 | $1054^{+7.9}_{-7.4}$ | $55^{+7.4}_{-12.6}$ | $-6^{+14.3}_{-12.7}$ | $-3^{+12.4}_{-8.3}$ | $0^{+14.1}_{-9.0}$ |
| 87 | zephyr-7b-alpha | $1051^{+24.4}_{-26.8}$ | $22^{+26.7}_{-28.7}$ | $-14^{+39.1}_{-37.1}$ | $-18^{+34.8}_{-25.8}$ | $-2^{+33.6}_{-30.5}$ |
| 88 | zephyr-7b-beta | $1049^{+11.4}_{-11.0}$ | $39^{+10.7}_{-16.5}$ | $-42^{+25.7}_{-24.8}$ | $-22^{+15.8}_{-11.5}$ | $-13^{+18.7}_{-12.9}$ |
| 89 | gemma-1.1-2b-it | $1044^{+9.0}_{-8.6}$ | $2^{+8.4}_{-15.2}$ | $9^{+16.3}_{-16.0}$ | $-19^{+16.0}_{-11.0}$ | $25^{+16.4}_{-12.3}$ |
| 90 | mpt-30b-chat | $1041^{+20.5}_{-21.3}$ | $31^{+20.8}_{-27.2}$ | $-14^{+38.9}_{-37.3}$ | $12^{+29.1}_{-22.5}$ | $-20^{+31.8}_{-30.0}$ |
| 91 | codellama-70b-instruct | $1039^{+24.2}_{-23.2}$ | $29^{+26.1}_{-30.3}$ | $25^{+36.5}_{-34.2}$ | $8^{+33.3}_{-28.6}$ | $-1^{+36.0}_{-28.9}$ |

Table 7: (continued)

| Rank | Model Name | Rating | English | Chinese | Hardness | Code |
|---|---|---|---|---|---|---|
| 92 | pplx-7b-online | $1038^{+15.2}_{-14.6}$ | $35^{+15.6}_{-21.0}$ | $21^{+32.7}_{-27.8}$ | $-17^{+21.6}_{-15.4}$ | $-21^{+22.7}_{-18.3}$ |
| 93 | llama-2-7b-chat | $1036^{+9.3}_{-8.6}$ | $45^{+7.9}_{-15.3}$ | $-28^{+19.0}_{-19.2}$ | $-22^{+14.4}_{-9.6}$ | $-31^{+16.4}_{-12.3}$ |
| 94 | guanaco-33b | $1035^{+20.7}_{-21.6}$ | $30^{+21.7}_{-25.9}$ | $-17^{+33.1}_{-31.7}$ | $-12^{+25.7}_{-21.4}$ | $-53^{+28.1}_{-27.0}$ |
| 95 | gemma-7b-it | $1029^{+11.3}_{-11.0}$ | $28^{+10.3}_{-16.5}$ | $38^{+18.7}_{-16.5}$ | $11^{+17.5}_{-11.6}$ | $7^{+16.7}_{-13.6}$ |
| 96 | stripedhyena-nous-7b | $1028^{+15.3}_{-13.6}$ | $21^{+14.1}_{-20.5}$ | $-16^{+35.2}_{-32.9}$ | $-19^{+19.2}_{-16.2}$ | $-10^{+24.2}_{-18.6}$ |
| 97 | qwen1.5-4b-chat | $1026^{+13.0}_{-10.7}$ | $-23^{+11.9}_{-18.2}$ | $36^{+19.6}_{-13.2}$ | $-13^{+17.2}_{-15.0}$ | $4^{+19.3}_{-15.4}$ |
| 98 | mistral-7b-instruct | $1008^{+11.9}_{-12.6}$ | $32^{+11.5}_{-18.0}$ | $-26^{+26.9}_{-27.6}$ | $-4^{+18.1}_{-11.6}$ | $0^{+19.2}_{-16.8}$ |
| 99 | palm-2 | $997^{+14.6}_{-14.4}$ | $43^{+14.4}_{-18.9}$ | $-69^{+29.4}_{-29.6}$ | $0^{+17.9}_{-13.2}$ | $-18^{+21.7}_{-16.6}$ |
| 100 | gemma-2b-it | $995^{+13.9}_{-14.7}$ | $20^{+15.2}_{-20.9}$ | $33^{+26.1}_{-24.6}$ | $-7^{+19.9}_{-17.1}$ | $7^{+25.0}_{-19.7}$ |
| 101 | olmo-7b-instruct | $995^{+13.4}_{-13.3}$ | $59^{+12.6}_{-18.3}$ | $54^{+18.8}_{-21.8}$ | $-30^{+18.8}_{-14.0}$ | $8^{+21.9}_{-18.7}$ |
| 102 | RWKV-4-Raven-14B | $971^{+17.6}_{-19.5}$ | $-30^{+17.5}_{-23.4}$ | $-28^{+30.0}_{-29.6}$ | $-23^{+23.6}_{-17.6}$ | $-7^{+24.5}_{-21.3}$ |
| 103 | koala-13b | $967^{+16.6}_{-17.4}$ | $31^{+16.6}_{-22.0}$ | $-44^{+25.5}_{-24.0}$ | $-36^{+19.5}_{-15.0}$ | $-5^{+21.8}_{-18.9}$ |
| 104 | alpaca-13b | $955^{+19.8}_{-17.7}$ | $-11^{+18.2}_{-24.9}$ | $-96^{+28.1}_{-28.4}$ | $-62^{+21.3}_{-15.3}$ | $-78^{+27.9}_{-21.8}$ |
| 105 | chatglm3-6b | $946^{+16.3}_{-17.0}$ | $33^{+17.3}_{-22.0}$ | $113^{+30.0}_{-32.1}$ | $4^{+21.8}_{-17.8}$ | $-7^{+25.3}_{-21.6}$ |
| 106 | mpt-7b-chat | $944^{+20.8}_{-20.3}$ | $8^{+20.6}_{-25.7}$ | $37^{+30.5}_{-30.1}$ | $-25^{+24.7}_{-22.6}$ | $-7^{+31.2}_{-26.0}$ |
| 107 | chatglm2-6b | $930^{+20.7}_{-19.2}$ | $30^{+21.2}_{-25.2}$ | $67^{+42.2}_{-37.8}$ | $-2^{+30.3}_{-24.8}$ | $-38^{+31.0}_{-27.1}$ |
| 108 | gpt4all-13b-snoozy | $924^{+25.1}_{-27.3}$ | $39^{+26.6}_{-29.2}$ | $-8^{+31.0}_{-31.9}$ | $10^{+28.9}_{-24.2}$ | $-21^{+33.1}_{-32.2}$ |
| 109 | oasst-pythia-12b | $912^{+17.6}_{-18.1}$ | $10^{+16.6}_{-21.2}$ | $-63^{+26.3}_{-25.1}$ | $-4^{+21.3}_{-18.4}$ | $-12^{+26.1}_{-20.0}$ |
| 110 | fastchat-t5-3b | $879^{+20.8}_{-19.6}$ | $42^{+18.6}_{-24.8}$ | $-108^{+28.6}_{-26.6}$ | $-36^{+23.6}_{-18.6}$ | $-90^{+28.8}_{-24.5}$ |
| 111 | chatglm-6b | $874^{+19.3}_{-19.1}$ | $11^{+18.1}_{-24.3}$ | $199^{+29.8}_{-28.7}$ | $18^{+22.4}_{-19.5}$ | $11^{+24.4}_{-23.1}$ |
| 112 | dolly-v2-12b | $856^{+22.7}_{-24.2}$ | $-15^{+21.9}_{-27.6}$ | $8^{+31.1}_{-30.9}$ | $4^{+26.2}_{-22.0}$ | $-58^{+30.8}_{-24.2}$ |
| 113 | llama-13b | $853^{+24.0}_{-25.8}$ | $-26^{+24.4}_{-28.6}$ | $-8^{+33.2}_{-27.7}$ | $-36^{+28.9}_{-27.0}$ | $-89^{+32.6}_{-25.1}$ |
| 114 | stablelm-tuned-alpha-7b | $843^{+19.6}_{-22.2}$ | $19^{+22.2}_{-24.7}$ | $30^{+29.1}_{-28.5}$ | $-13^{+24.0}_{-20.5}$ | $32^{+27.9}_{-23.9}$ |

Table 8: Leaderboard of human evaluation with modifiers when all fitted separately using a uni-dimensional approach. For the four tasks, modifiers are shown to indicate the deviation from the main rating to make comparison with POLYRATING easier. Indicated deviations are 95% confidence intervals determined using bootstrapping.

| Rank | Model Name | Rating | English | Chinese | Hardness | Code |
|---|---|---|---|---|---|---|
| 1 | gpt-4o-2024-05-13 | $1283^{+2.7}_{-2.8}$ | $-19^{+4.3}_{-4.4}$ | $52^{+9.0}_{-9.2}$ | $5^{+5.6}_{-5.4}$ | $13^{+6.3}_{-6.7}$ |
| 2 | claude-3-5-sonnet-20240620 | $1267^{+4.1}_{-4.2}$ | $-23^{+6.9}_{-6.9}$ | $45^{+12.1}_{-12.1}$ | $11^{+8.5}_{-8.8}$ | $35^{+9.1}_{-10.0}$ |
| 3 | gemini-advanced-0514 | $1261^{+2.8}_{-2.9}$ | $-28^{+4.8}_{-5.1}$ | $69^{+8.7}_{-8.9}$ | $-8^{+6.2}_{-6.1}$ | $-4^{+7.0}_{-7.5}$ |
| 4 | gemini-1.5-pro-api-0514 | $1259^{+2.7}_{-2.9}$ | $-25^{+4.8}_{-4.9}$ | $76^{+8.5}_{-8.7}$ | $5^{+6.1}_{-5.8}$ | $8^{+6.8}_{-7.0}$ |
| 5 | gpt-4-turbo-2024-04-09 | $1252^{+2.3}_{-2.3}$ | $-12^{+3.7}_{-4.0}$ | $50^{+7.4}_{-7.1}$ | $5^{+5.1}_{-4.7}$ | $14^{+6.0}_{-5.8}$ |
| 6 | gpt-4-1106-preview | $1248^{+2.3}_{-2.3}$ | $-17^{+3.4}_{-3.9}$ | $52^{+7.5}_{-7.9}$ | $0^{+4.9}_{-5.2}$ | $8^{+5.6}_{-5.9}$ |
| 7 | gemini-1.5-pro-api-0409-preview | $1247^{+2.6}_{-2.6}$ | $-16^{+4.3}_{-4.0}$ | $55^{+8.3}_{-8.5}$ | $-15^{+5.5}_{-5.8}$ | $-15^{+6.3}_{-6.0}$ |
| 8 | claude-3-opus-20240229 | $1245^{+2.1}_{-2.1}$ | $-31^{+3.4}_{-3.4}$ | $70^{+6.6}_{-6.6}$ | $3^{+4.3}_{-4.3}$ | $6^{+5.1}_{-5.3}$ |
| 9 | gpt-4-0125-preview | $1243^{+2.2}_{-2.4}$ | $-16^{+3.7}_{-3.6}$ | $53^{+7.2}_{-7.4}$ | $2^{+5.1}_{-5.0}$ | $3^{+6.0}_{-6.2}$ |
| 10 | yi-large-preview | $1233^{+2.8}_{-2.8}$ | $-19^{+4.7}_{-4.4}$ | $84^{+8.9}_{-8.8}$ | $7^{+6.1}_{-6.0}$ | $11^{+6.7}_{-7.1}$ |
| 11 | gemini-1.5-flash-api-0514 | $1226^{+3.0}_{-3.0}$ | $-25^{+5.0}_{-4.7}$ | $66^{+9.4}_{-9.3}$ | $1^{+6.0}_{-6.3}$ | $9^{+7.3}_{-7.3}$ |
| 12 | yi-large | $1215^{+4.4}_{-4.5}$ | $-16^{+8.3}_{-8.2}$ | $58^{+15.3}_{-15.1}$ | $6^{+10.2}_{-10.2}$ | $18^{+11.9}_{-11.9}$ |
| 13 | gemma-2-27b-it | $1210^{+5.9}_{-6.0}$ | $-20^{+10.1}_{-10.0}$ | $54^{+20.1}_{-20.1}$ | $-18^{+11.7}_{-12.5}$ | $-5^{+14.6}_{-14.0}$ |
| 14 | bard-jan-24-gemini-pro | $1207^{+4.7}_{-4.5}$ | $-25^{+7.0}_{-7.0}$ | $59^{+24.1}_{-22.5}$ | $-51^{+10.6}_{-10.6}$ | $-35^{+13.1}_{-13.3}$ |
| 15 | glm-4-0520 | $1206^{+5.0}_{-5.1}$ | $-15^{+8.2}_{-8.7}$ | $90^{+17.2}_{-16.0}$ | $10^{+10.7}_{-10.9}$ | $13^{+12.7}_{-12.4}$ |
| 16 | nemotron-4-340b-instruct | $1204^{+3.6}_{-3.9}$ | $-22^{+6.4}_{-6.4}$ | $62^{+12.3}_{-12.4}$ | $-4^{+8.3}_{-8.2}$ | $-7^{+11.2}_{-9.9}$ |
| 17 | llama-3-70b-instruct | $1202^{+2.1}_{-2.1}$ | $22^{+3.3}_{-3.4}$ | $-32^{+6.5}_{-6.6}$ | $-5^{+4.5}_{-4.5}$ | $0^{+5.3}_{-5.3}$ |
| 18 | claude-3-sonnet-20240229 | $1198^{+2.2}_{-2.2}$ | $-23^{+3.5}_{-3.4}$ | $48^{+6.1}_{-6.6}$ | $1^{+4.4}_{-4.7}$ | $17^{+5.4}_{-5.1}$ |
| 19 | reka-core-20240501 | $1194^{+2.4}_{-2.4}$ | $-20^{+4.0}_{-4.3}$ | $63^{+8.3}_{-8.5}$ | $-6^{+5.2}_{-5.8}$ | $-4^{+6.2}_{-5.9}$ |

Table 8: (continued)

| Rank | Model Name | Rating | English | Chinese | Hardness | Code |
|------|------------|--------|---------|---------|----------|------|
| 20 | command-r-plus | $1188^{+2.2}_{-2.3}$ | $-25^{+3.6}_{-3.7}$ | $63^{+6.9}_{-7.0}$ | $-20^{+5.1}_{-4.8}$ | $-21^{+5.9}_{-5.9}$ |
| 21 | gpt-4-0314 | $1188^{+2.5}_{-2.6}$ | $-23^{+4.2}_{-4.1}$ | $54^{+8.3}_{-8.3}$ | $9^{+5.5}_{-5.6}$ | $10^{+6.5}_{-6.7}$ |
| 22 | qwen-max-0428 | $1185^{+2.7}_{-2.9}$ | $-17^{+5.1}_{-4.9}$ | $105^{+11.3}_{-11.4}$ | $3^{+7.1}_{-7.1}$ | $7^{+7.8}_{-8.3}$ |
| 23 | qwen2-72b-instruct | $1184^{+3.3}_{-3.2}$ | $-17^{+5.5}_{-5.3}$ | $106^{+10.6}_{-10.4}$ | $1^{+7.2}_{-6.9}$ | $-1^{+8.6}_{-8.6}$ |
| 24 | claude-3-haiku-20240307 | $1181^{+2.2}_{-2.2}$ | $-23^{+3.6}_{-3.8}$ | $33^{+6.4}_{-7.0}$ | $1^{+4.7}_{-4.6}$ | $10^{+5.6}_{-5.7}$ |
| 25 | gemma-2-9b-it | $1180^{+6.0}_{-6.0}$ | $-24^{+10.5}_{-10.1}$ | $56^{+19.9}_{-19.5}$ | $-10^{+12.7}_{-13.3}$ | $-17^{+14.8}_{-15.1}$ |
| 26 | deepseek-coder-v2 | $1177^{+4.6}_{-4.9}$ | $-28^{+8.3}_{-8.1}$ | $80^{+16.2}_{-16.2}$ | $37^{+10.3}_{-10.8}$ | $61^{+11.4}_{-11.8}$ |
| 27 | glm-4-0116 | $1173^{+5.1}_{-5.3}$ | $2^{+9.0}_{-9.1}$ | $99^{+18.5}_{-17.9}$ | $14^{+10.6}_{-11.3}$ | $19^{+12.7}_{-13.0}$ |
| 28 | qwen1.5-110b-chat | $1165^{+3.1}_{-3.1}$ | $-13^{+4.8}_{-5.1}$ | $96^{+10.4}_{-10.1}$ | $4^{+6.3}_{-6.4}$ | $10^{+8.1}_{-7.9}$ |
| 29 | gpt-4-0613 | $1165^{+2.2}_{-2.4}$ | $-18^{+3.5}_{-3.8}$ | $25^{+7.6}_{-7.8}$ | $3^{+4.8}_{-5.0}$ | $3^{+5.7}_{-6.0}$ |
| 30 | reka-flash-preview-20240611 | $1164^{+4.3}_{-4.1}$ | $-22^{+7.7}_{-7.7}$ | $52^{+13.4}_{-13.5}$ | $-14^{+8.6}_{-8.5}$ | $-3^{+10.1}_{-9.8}$ |
| 31 | yi-1.5-34b-chat | $1158^{+3.3}_{-3.4}$ | $5^{+5.7}_{-5.8}$ | $110^{+11.0}_{-11.1}$ | $2^{+7.0}_{-8.0}$ | $6^{+8.7}_{-9.0}$ |
| 32 | reka-flash-21b-20240226-online | $1155^{+3.6}_{-3.7}$ | $-19^{+6.1}_{-6.7}$ | $46^{+11.0}_{-10.9}$ | $-12^{+8.5}_{-8.8}$ | $-6^{+8.4}_{-8.9}$ |
| 33 | mistral-large-2402 | $1155^{+2.5}_{-2.4}$ | $-9^{+4.0}_{-4.2}$ | $18^{+7.9}_{-7.8}$ | $10^{+5.5}_{-5.3}$ | $14^{+6.2}_{-6.2}$ |
| 34 | llama-3-8b-instruct | $1154^{+2.2}_{-2.3}$ | $11^{+3.6}_{-3.7}$ | $-15^{+6.7}_{-6.8}$ | $-16^{+4.6}_{-4.6}$ | $-6^{+5.8}_{-5.5}$ |
| 35 | qwen1.5-72b-chat | $1151^{+2.4}_{-2.6}$ | $-16^{+4.3}_{-4.3}$ | $94^{+9.0}_{-9.1}$ | $-4^{+5.8}_{-5.9}$ | $11^{+6.8}_{-7.4}$ |
| 36 | claude-1 | $1150^{+3.7}_{-3.8}$ | $-21^{+14.4}_{-5.8}$ | $49^{+14.4}_{-13.7}$ | $-25^{+7.9}_{-8.5}$ | $-12^{+10.5}_{-10.4}$ |
| 37 | command-r | $1149^{+2.5}_{-2.5}$ | $-26^{+4.2}_{-4.3}$ | $66^{+7.2}_{-7.5}$ | $-30^{+5.5}_{-5.6}$ | $-23^{+6.4}_{-6.0}$ |
| 38 | reka-flash-21b-20240226 | $1147^{+3.1}_{-3.3}$ | $-20^{+5.1}_{-5.1}$ | $46^{+10.2}_{-9.9}$ | $-13^{+7.1}_{-6.3}$ | $-4^{+7.3}_{-8.1}$ |
| 39 | mistral-medium | $1146^{+2.9}_{-2.8}$ | $-9^{+4.7}_{-4.9}$ | $20^{+9.9}_{-10.4}$ | $-1^{+6.0}_{-6.3}$ | $8^{+7.6}_{-7.2}$ |
| 40 | mixtral-8x22b-instruct-v0.1 | $1146^{+2.7}_{-2.7}$ | $-10^{+4.4}_{-4.5}$ | $40^{+8.5}_{-8.5}$ | $3^{+5.9}_{-6.2}$ | $9^{+6.4}_{-6.9}$ |
| 41 | gemini-pro-dev-api | $1136^{+3.6}_{-3.7}$ | $-26^{+6.3}_{-5.8}$ | $51^{+12.6}_{-12.6}$ | $-27^{+8.8}_{-8.9}$ | $-31^{+10.0}_{-10.0}$ |
| 42 | claude-2.0 | $1133^{+4.7}_{-4.5}$ | $-19^{+6.9}_{-6.8}$ | $58^{+21.8}_{-21.0}$ | $-5^{+10.6}_{-10.1}$ | $4^{+13.5}_{-13.1}$ |
| 43 | qwen1.5-32b-chat | $1132^{+3.3}_{-3.4}$ | $-21^{+5.5}_{-5.2}$ | $104^{+9.6}_{-9.8}$ | $4^{+7.5}_{-7.4}$ | $16^{+8.6}_{-8.7}$ |
| 44 | zephyr-orpo-141b-A35b-v0.1 | $1128^{+6.3}_{-6.0}$ | $-9^{+11.4}_{-10.6}$ | $22^{+18.8}_{-18.1}$ | $-11^{+14.5}_{-15.1}$ | $-5^{+16.6}_{-17.2}$ |
| 45 | mistral-next | $1127^{+4.5}_{-4.5}$ | $-16^{+6.5}_{-6.9}$ | $12^{+17.2}_{-17.8}$ | $3^{+9.8}_{-9.1}$ | $7^{+12.1}_{-11.8}$ |
| 46 | phi-3-medium-4k-instruct | $1125^{+4.1}_{-4.1}$ | $-7^{+7.0}_{-6.6}$ | $36^{+12.2}_{-11.9}$ | $9^{+8.8}_{-9.0}$ | $10^{+9.9}_{-9.2}$ |
| 47 | gpt-3.5-turbo-0613 | $1120^{+2.9}_{-2.9}$ | $-21^{+4.5}_{-4.5}$ | $40^{+13.5}_{-13.3}$ | $4^{+6.8}_{-6.6}$ | $16^{+8.6}_{-8.3}$ |
| 48 | qwen1.5-14b-chat | $1119^{+3.4}_{-3.7}$ | $-19^{+6.0}_{-6.1}$ | $89^{+9.8}_{-9.7}$ | $1^{+7.4}_{-7.4}$ | $9^{+8.8}_{-9.1}$ |
| 49 | starling-lm-7b-beta | $1119^{+3.7}_{-4.0}$ | $-12^{+6.4}_{-6.3}$ | $66^{+10.5}_{-10.3}$ | $-1^{+7.8}_{-8.1}$ | $13^{+9.1}_{-9.2}$ |
| 50 | claude-2.1 | $1118^{+3.0}_{-2.9}$ | $-22^{+4.6}_{-4.5}$ | $32^{+10.9}_{-11.8}$ | $2^{+6.4}_{-6.5}$ | $15^{+8.0}_{-8.7}$ |
| 51 | yi-34b-chat | $1116^{+3.9}_{-3.8}$ | $-9^{+6.6}_{-6.3}$ | $116^{+14.7}_{-14.3}$ | $-12^{+8.8}_{-9.2}$ | $-9^{+10.5}_{-10.7}$ |
| 52 | gemini-pro | $1115^{+6.2}_{-6.1}$ | $-16^{+9.6}_{-9.4}$ | $52^{+29.4}_{-30.2}$ | $-30^{+14.4}_{-13.5}$ | $-19^{+17.3}_{-19.3}$ |
| 53 | mixtral-8x7b-instruct-v0.1 | $1114^{+0.0}_{0.0}$ | $0^{+0.0}_{0.0}$ | $0^{+0.0}_{0.0}$ | $0^{+0.0}_{0.0}$ | $0^{+0.0}_{0.0}$ |
| 54 | gpt-3.5-turbo-0125 | $1113^{+2.4}_{-2.2}$ | $-24^{+4.0}_{-4.0}$ | $29^{+7.6}_{-7.7}$ | $1^{+5.4}_{-5.1}$ | $12^{+6.1}_{-6.5}$ |
| 55 | claude-instant-1 | $1112^{+3.9}_{-3.9}$ | $-16^{+6.1}_{-5.8}$ | $38^{+17.5}_{-17.2}$ | $-4^{+8.7}_{-8.7}$ | $-2^{+11.3}_{-10.8}$ |
| 56 | wizardlm-70b | $1110^{+5.4}_{-5.6}$ | $-11^{+8.2}_{-8.7}$ | $10^{+27.5}_{-27.7}$ | $-31^{+12.9}_{-12.2}$ | $-38^{+15.7}_{-15.3}$ |
| 57 | gpt-3.5-turbo-0314 | $1110^{+7.3}_{-7.2}$ | $-26^{+11.2}_{-11.8}$ | $95^{+30.4}_{-30.5}$ | $9^{+18.4}_{-20.6}$ | $8^{+24.3}_{-24.1}$ |
| 58 | dbrx-instruct-preview | $1106^{+2.8}_{-3.1}$ | $-2^{+4.7}_{-4.7}$ | $30^{+8.2}_{-8.5}$ | $2^{+6.2}_{-6.4}$ | $15^{+7.4}_{-7.4}$ |
| 59 | phi-3-small-8k-instruct | $1104^{+3.9}_{-3.8}$ | $2^{+6.9}_{-6.7}$ | $21^{+11.7}_{-11.6}$ | $3^{+8.6}_{-8.1}$ | $3^{+9.4}_{-10.0}$ |
| 60 | tulu-2-dpo-70b | $1103^{+6.2}_{-6.1}$ | $-11^{+8.5}_{-9.3}$ | $-33^{+29.0}_{-31.3}$ | $-5^{+14.3}_{-13.9}$ | $-8^{+17.0}_{-18.0}$ |
| 61 | snowflake-arctic-instruct | $1099^{+2.9}_{-2.8}$ | $-20^{+4.7}_{-4.6}$ | $53^{+9.2}_{-9.6}$ | $-23^{+6.2}_{-6.1}$ | $-19^{+7.3}_{-8.1}$ |
| 62 | openchat-3.5-0106 | $1099^{+4.2}_{-4.1}$ | $-15^{+6.3}_{-6.7}$ | $48^{+14.0}_{-13.0}$ | $-13^{+8.7}_{-8.5}$ | $6^{+11.5}_{-11.4}$ |
| 63 | llama-2-70b-chat | $1097^{+2.7}_{-2.8}$ | $1^{+4.4}_{-4.7}$ | $-39^{+9.5}_{-9.4}$ | $-25^{+6.0}_{-5.8}$ | $-22^{+7.7}_{-7.7}$ |
| 64 | vicuna-33b | $1095^{+3.6}_{-3.4}$ | $-7^{+5.2}_{-5.5}$ | $9^{+13.1}_{-12.9}$ | $-27^{+7.8}_{-8.2}$ | $-28^{+9.8}_{-9.7}$ |
| 65 | starling-lm-7b-alpha | $1093^{+4.9}_{-4.7}$ | $-7^{+7.4}_{-7.8}$ | $20^{+17.5}_{-17.1}$ | $-17^{+10.8}_{-10.6}$ | $-10^{+12.6}_{-12.9}$ |
| 66 | gemma-1.1-7b-it | $1090^{+3.3}_{-3.3}$ | $-8^{+5.4}_{-5.4}$ | $39^{+9.9}_{-9.3}$ | $-13^{+7.2}_{-7.1}$ | $-3^{+7.8}_{-8.0}$ |
| 67 | nous-hermes-2-mixtral-8x7b-dpo | $1087^{+7.6}_{-6.8}$ | $-9^{+11.4}_{-11.4}$ | $-9^{+47.6}_{-48.2}$ | $-35^{+15.8}_{-15.5}$ | $-4^{+18.7}_{-19.9}$ |
| 68 | llama2-70b-steerlm-chat | $1083^{+7.9}_{-8.5}$ | $-14^{+12.9}_{-12.6}$ | $14^{+37.5}_{-35.5}$ | $-35^{+19.9}_{-20.0}$ | $-59^{+23.2}_{-23.0}$ |
| 69 | openchat-3.5 | $1080^{+5.4}_{-5.3}$ | $-14^{+7.8}_{-8.7}$ | $53^{+26.8}_{-25.7}$ | $-14^{+12.1}_{-12.7}$ | $-24^{+15.4}_{-16.3}$ |

Table 8: (continued)

| Rank | Model Name | Rating | English | Chinese | Hardness | Code |
|---|---|---|---|---|---|---|
| 70 | deepseek-llm-67b-chat | $1080^{+7.0}_{-7.0}$ | $-15^{+10.3}_{-10.8}$ | $103^{+34.8}_{-33.8}$ | $-14^{+15.3}_{-15.5}$ | $2^{+20.1}_{-19.7}$ |
| 71 | openhermes-2.5-mistral-7b | $1080^{+6.4}_{-6.3}$ | $-6^{+9.4}_{-9.9}$ | $8^{+34.6}_{-33.3}$ | $-10^{+15.1}_{-14.8}$ | $-18^{+20.2}_{-19.9}$ |
| 72 | qwen1.5-7b-chat | $1079^{+6.3}_{-6.3}$ | $-22^{+10.1}_{-10.1}$ | $122^{+22.7}_{-23.3}$ | $-10^{+12.9}_{-13.7}$ | $13^{+18.0}_{-17.7}$ |
| 73 | pplx-70b-online | $1077^{+5.9}_{-6.1}$ | $-12^{+9.2}_{-9.4}$ | $54^{+29.6}_{-30.5}$ | $-49^{+13.4}_{-13.7}$ | $-49^{+16.0}_{-15.5}$ |
| 74 | mistral-7b-instruct-v0.2 | $1075^{+3.7}_{-3.5}$ | $7^{+5.7}_{-5.7}$ | $8^{+11.2}_{-10.7}$ | $-6^{+7.9}_{-7.4}$ | $0^{+9.2}_{-9.6}$ |
| 75 | gpt-3.5-turbo-1106 | $1073^{+3.9}_{-3.8}$ | $-18^{+6.2}_{-6.1}$ | $10^{+17.8}_{-19.0}$ | $20^{+9.0}_{-9.3}$ | $25^{+10.8}_{-10.3}$ |
| 76 | phi-3-mini-4k-instruct | $1072^{+3.7}_{-3.7}$ | $3^{+6.1}_{-6.1}$ | $12^{+12.0}_{-11.5}$ | $8^{+7.1}_{-7.1}$ | $16^{+8.4}_{-8.7}$ |
| 77 | llama-2-13b-chat | $1068^{+4.0}_{-3.8}$ | $-6^{+5.8}_{-6.3}$ | $-17^{+15.5}_{-15.4}$ | $-19^{+9.0}_{-8.7}$ | $-16^{+10.5}_{-10.8}$ |
| 78 | solar-10.7b-instruct-v1.0 | $1067^{+7.2}_{-7.2}$ | $-5^{+10.4}_{-11.1}$ | $0^{+37.2}_{-35.9}$ | $-9^{+17.0}_{-16.7}$ | $-16^{+19.8}_{-21.1}$ |
| 79 | dolphin-2.2.1-mistral-7b | $1066^{+11.4}_{-11.3}$ | $-5^{+16.6}_{-17.2}$ | $44^{+56.3}_{-54.6}$ | $-15^{+27.3}_{-26.6}$ | $-40^{+34.2}_{-34.6}$ |
| 80 | wizardlm-13b | $1063^{+6.0}_{-5.6}$ | $-12^{+8.9}_{-8.7}$ | $28^{+26.1}_{-26.0}$ | $-49^{+15.0}_{-14.3}$ | $-37^{+18.8}_{-17.3}$ |
| 81 | zephyr-7b-beta | $1057^{+5.2}_{-5.0}$ | $-2^{+7.6}_{-7.1}$ | $-31^{+24.9}_{-24.3}$ | $-29^{+11.9}_{-11.5}$ | $-24^{+14.2}_{-13.8}$ |
| 82 | phi-3-mini-128k-instruct | $1054^{+3.6}_{-3.4}$ | $-15^{+5.7}_{-5.4}$ | $47^{+11.1}_{-11.4}$ | $-15^{+7.6}_{-7.4}$ | $-23^{+8.7}_{-9.2}$ |
| 83 | vicuna-13b | $1050^{+4.2}_{-3.9}$ | $-19^{+6.2}_{-6.2}$ | $63^{+15.5}_{-16.3}$ | $-23^{+9.2}_{-8.9}$ | $-15^{+11.6}_{-11.8}$ |
| 84 | mpt-30b-chat | $1050^{+9.4}_{-9.4}$ | $-5^{+14.0}_{-15.1}$ | $-3^{+48.4}_{-48.3}$ | $-2^{+25.1}_{-25.0}$ | $-21^{+28.5}_{-30.3}$ |
| 85 | codellama-34b-instruct | $1049^{+6.0}_{-5.7}$ | $-13^{+8.5}_{-8.5}$ | $-17^{+14.5}_{-29.4}$ | $-19^{+14.5}_{-13.2}$ | $-8^{+16.9}_{-17.2}$ |
| 86 | zephyr-7b-alpha | $1048^{+10.7}_{-10.9}$ | $-6^{+16.1}_{-16.4}$ | $1^{+68.5}_{-68.4}$ | $-27^{+28.2}_{-28.5}$ | $-14^{+31.9}_{-32.8}$ |
| 87 | codellama-70b-instruct | $1047^{+14.6}_{-13.5}$ | $-3^{+22.8}_{-22.0}$ | $62^{+39.8}_{-37.2}$ | $3^{+29.9}_{-28.6}$ | $2^{+34.8}_{-36.9}$ |
| 88 | pplx-7b-online | $1045^{+6.4}_{-6.6}$ | $-6^{+10.2}_{-9.6}$ | $47^{+33.3}_{-31.8}$ | $-28^{+14.7}_{-14.4}$ | $-30^{+17.8}_{-17.1}$ |
| 89 | gemma-7b-it | $1043^{+5.3}_{-5.2}$ | $-8^{+8.3}_{-8.0}$ | $64^{+16.0}_{-15.6}$ | $4^{+11.8}_{-11.4}$ | $7^{+13.7}_{-12.8}$ |
| 90 | llama-2-7b-chat | $1043^{+4.1}_{-4.1}$ | $1^{+6.3}_{-6.5}$ | $-11^{+15.9}_{-15.6}$ | $-32^{+9.8}_{-9.2}$ | $-38^{+11.3}_{-11.7}$ |
| 91 | qwen-14b-chat | $1041^{+6.8}_{-6.4}$ | $-21^{+10.3}_{-10.6}$ | $94^{+34.7}_{-36.5}$ | $-16^{+16.3}_{-16.8}$ | $16^{+19.6}_{-20.5}$ |
| 92 | falcon-180b-chat | $1039^{+14.1}_{-13.3}$ | $-12^{+20.6}_{-21.2}$ | $-6^{+79.9}_{-103.1}$ | $-38^{+32.2}_{-31.7}$ | $-22^{+42.0}_{-43.5}$ |
| 93 | guanaco-33b | $1037^{+9.1}_{-9.4}$ | $-7^{+13.0}_{-13.7}$ | $-8^{+37.5}_{-37.8}$ | $-35^{+23.6}_{-23.0}$ | $-69^{+28.3}_{-27.9}$ |
| 94 | gemma-1.1-2b-it | $1034^{+4.5}_{-4.7}$ | $-15^{+13.9}_{-8.0}$ | $54^{+13.9}_{-13.8}$ | $-15^{+10.2}_{-10.6}$ | $8^{+11.5}_{-11.1}$ |
| 95 | stripedhyena-nous-7b | $1024^{+6.8}_{-6.8}$ | $-8^{+9.9}_{-9.9}$ | $11^{+36.3}_{-36.7}$ | $-26^{+14.5}_{-14.3}$ | $-21^{+18.8}_{-17.9}$ |
| 96 | olmo-7b-instruct | $1021^{+6.1}_{-6.0}$ | $0^{+9.9}_{-9.7}$ | $62^{+18.4}_{-18.5}$ | $-26^{+12.5}_{-13.2}$ | $-5^{+17.3}_{-16.3}$ |
| 97 | mistral-7b-instruct | $1016^{+5.6}_{-5.3}$ | $-4^{+8.1}_{-8.4}$ | $-8^{+26.1}_{-25.7}$ | $-10^{+11.6}_{-12.6}$ | $-5^{+15.3}_{-14.5}$ |
| 98 | palm-2 | $1012^{+5.7}_{-5.9}$ | $-1^{+8.7}_{-8.2}$ | $-69^{+30.6}_{-32.4}$ | $-12^{+14.4}_{-13.9}$ | $-22^{+16.5}_{-16.6}$ |
| 99 | vicuna-7b | $1012^{+6.2}_{-6.2}$ | $-20^{+9.4}_{-9.2}$ | $30^{+25.6}_{-27.8}$ | $-20^{+15.6}_{-15.3}$ | $-27^{+17.8}_{-17.3}$ |
| 100 | qwen1.5-4b-chat | $1003^{+5.5}_{-5.3}$ | $-30^{+15.8}_{-9.0}$ | $92^{+15.8}_{-16.5}$ | $-21^{+11.8}_{-11.5}$ | $-11^{+15.6}_{-14.3}$ |
| 101 | gemma-2b-it | $1000^{+6.7}_{-6.8}$ | $-11^{+10.7}_{-11.2}$ | $67^{+20.8}_{-21.3}$ | $-12^{+14.8}_{-14.9}$ | $0^{+18.2}_{-18.4}$ |
| 102 | koala-13b | $971^{+6.5}_{-6.6}$ | $-6^{+9.9}_{-9.5}$ | $-34^{+25.9}_{-16.4}$ | $-43^{+16.8}_{-16.4}$ | $-30^{+18.7}_{-18.9}$ |
| 103 | chatglm3-6b | $962^{+7.9}_{-7.7}$ | $-11^{+11.6}_{-11.2}$ | $159^{+33.1}_{-35.2}$ | $-7^{+17.2}_{-17.0}$ | $-9^{+22.6}_{-22.3}$ |
| 104 | gpt4all-13b-snoozy | $941^{+11.6}_{-12.2}$ | $-4^{+18.2}_{-17.0}$ | $-6^{+48.8}_{-44.9}$ | $-6^{+28.2}_{-28.5}$ | $-28^{+35.3}_{-37.5}$ |
| 105 | chatglm2-6b | $936^{+9.9}_{-9.8}$ | $-7^{+14.9}_{-15.0}$ | $120^{+47.3}_{-46.3}$ | $-19^{+25.1}_{-23.1}$ | $-43^{+29.9}_{-30.5}$ |
| 106 | mpt-7b-chat | $935^{+8.0}_{-8.2}$ | $-15^{+11.4}_{-11.9}$ | $73^{+34.0}_{-32.0}$ | $-36^{+20.6}_{-18.5}$ | $-28^{+26.1}_{-24.7}$ |
| 107 | RWKV-4-Raven-14B | $929^{+8.0}_{-7.7}$ | $-19^{+11.4}_{-11.5}$ | $29^{+31.6}_{-30.9}$ | $-36^{+18.2}_{-18.2}$ | $-30^{+23.7}_{-22.2}$ |
| 108 | alpaca-13b | $911^{+7.0}_{-7.2}$ | $-13^{+10.2}_{-10.1}$ | $-65^{+30.4}_{-30.9}$ | $-88^{+18.0}_{-17.7}$ | $-117^{+22.7}_{-24.0}$ |
| 109 | oasst-pythia-12b | $902^{+7.0}_{-6.8}$ | $-9^{+11.5}_{-10.9}$ | $-41^{+26.8}_{-26.6}$ | $-18^{+18.1}_{-16.4}$ | $-26^{+22.4}_{-21.6}$ |
| 110 | chatglm-6b | $889^{+7.9}_{-7.5}$ | $-25^{+10.8}_{-11.6}$ | $252^{+30.0}_{-30.3}$ | $9^{+19.0}_{-19.4}$ | $1^{+22.5}_{-23.3}$ |
| 111 | fastchat-t5-3b | $879^{+7.9}_{-7.9}$ | $-1^{+11.7}_{-12.0}$ | $22^{+172.3}_{-250.3}$ | $-66^{+18.6}_{-19.0}$ | $-117^{+25.8}_{-25.2}$ |
| 112 | stablelm-tuned-alpha-7b | $851^{+9.7}_{-9.2}$ | $-13^{+13.3}_{-14.1}$ | $52^{+35.9}_{-31.8}$ | $-17^{+23.8}_{-22.6}$ | $10^{+27.3}_{-26.3}$ |
| 113 | dolly-v2-12b | $828^{+9.2}_{-9.3}$ | $-20^{+14.0}_{-13.5}$ | $53^{+39.2}_{-35.2}$ | $-27^{+23.2}_{-22.3}$ | $-78^{+27.3}_{-28.5}$ |
| 114 | llama-13b | $806^{+10.2}_{-10.6}$ | $-23^{+16.6}_{-16.7}$ | $47^{+35.7}_{-36.2}$ | $-79^{+28.7}_{-27.8}$ | $-134^{+35.4}_{-35.6}$ |

