# OpenReview forum: "Polyrating: A Cost-Effective and Bias-Aware Rating System for LLM Evaluation"
_ICLR.cc/2025/Conference — ICLR 2025 Poster_

### Official Review · Reviewer_1REa · 2024-10-26

**Soundness:** 3
**Presentation:** 2
**Contribution:** 2
**Rating:** 6
**Confidence:** 4

**Summary:**

This manuscript introduces Polyrating, a multivariate rating system for evaluating Large Language Models (LLMs). The authors aim to address several key challenges in current LLM evaluation methods, including accounting for biases, reducing the cost of human evaluations, and enabling meaningful comparisons of model performance across different tasks. Polyrating uses maximum a posteriori (MAP) estimation to incorporate different tasks and biases into the rating process. Experiments are conducted on publicly available human preference data to demonstrate the effectiveness of the proposed system.

**Strengths:**

1. The authors clearly identify important limitations in existing LLM evaluation approaches, specifically focusing on the influence of biases, the high cost of human evaluation, and the lack of comparability across tasks. The proposed Polyrating is designed to directly address these challenges, providing a clear motivation for the work.

2. The manuscript provides a theoretical analysis of Polyrating, demonstrating its convergence properties. This provides a strong theoretical foundation for the proposed system and strengthens the confidence in its practical applicability.

**Weaknesses:**

1. The manuscript lacks a comparison with existing multivariate rating systems. Empirical comparison and discussion of adaptations for LLMs are needed.

2. Experimental evidence to support some of the claimed benefits of Polyrating, is insufficient. Model selection for task comparability is questionable. The efficiency analysis lacks clarity on calculation methods.

**Questions:**

1. Line 233: Why are different optimization methods (Newton's method and L-BFGS) used for different parameters? What's the rationale?

2. How are sample efficiency improvements (e.g., "38%" in line 361) calculated?

3. In Table 2 and 3, several models are chosen to illustrate the differences in task comparability between Polyrating and a unidimensional approach. The following questions arise:

3-1: Line 410: What specifically makes Mixtral's performance "very different"? The tables don't clearly show this. Providing a more precise explanation or visualization of these differences is necessary to support this claim.

3-2: Line 413: The Chinese task analysis uses proprietary models with undisclosed training details and the Mixtral model, which, due to its specific role in the Chatbot Arena setup, is not suitable for this comparison. Please consider revising the experimental setup to enhance the validity of the comparisons.

4. While existing multivariate rating systems are mentioned, their performance isn't compared to Polyrating. Include empirical comparisons and highlight performance differences would significantly strengthen the manuscript.

---

> ### Author Response · Authors · 2024-11-18
>
> We thank the reviewer for their detailed review and insightful questions. We are happy to hear that they find our work provides a solid motivation which Polyrating directly addresses and that they appreciate our theoretical analysis. Below, we address their remaining questions.
>
> **Why did you not include other multivariate rating systems in your experiments?**
> We did not include other multivariate rating systems in our experiments because they fundamentally differ from Polyrating in how they handle bias and model-specific parameters, making them categorically unsuitable for the experimental results we obtain.
>
> Traditional multivariate rating systems are all based on approximative algorithms like Elo and Glicko, which were designed to handle time-dependent ratings of millions of objects or people. Thus, relying on these rating systems to obtain task-specific ratings would significantly reduce effectiveness compared to the optimal MLE-based ratings. Fig. 4 in Appendix A shows just how poor these approximations work in our LLM-based setting: the loss associated with Glicko (an improved version of Elo) is more than an order of magnitude higher compared to the MLE-based ratings! In contrast, Polyrating provably converges to the same ratings as the MLE-based one, not losing any utility in the rating. Furthermore, the implementations of these traditional multivariate rating systems require non-trivial adjustments to apply in the LLM-based setting due to their time dependence and various other differences that do not translate.
>
> Length-Controlled Alpaca-Eval is the only system based on the Bradley-Terry (BT) model like Polyrating that can also incorporate various factors in its modeling. However, it misses several key features which make it inapplicable for the experiments we performed. First, it only includes model-pair-specific parameters without model-agnostic or model-specific parameters. This makes it categorically impossible to obtain task-specific ratings with it. This restriction also makes it impossible to arrive at single-parameter estimates for the overall bias impact, a core requirement for our bias detection experiments. Furthermore, it does not contain any priors on parameters. This means that even if it were adjusted to incorporate model-specific parameters, it would still be *mathematically equivalent* to run the univariate baseline as there is no coupling between the rating of two tasks.
>
> We have now clarified these points in our Related Work section.
>
> **How are sample efficiency improvements (e.g., "38%" in line 361) calculated?**
> We measure this efficiency improvement by first obtaining the loss associated with Polyrating ratings at 10,000 samples. We then compute the lowest number of samples the univariate baseline requires to match this loss. For instance, in the case of 38%, the univariate case requires 16,130 samples to obtain the same loss since $(100 - 38)\\% \cdot 16,130 = 10,000$. Generally, for less than 10,000 samples this efficiency improvement would be larger, while for more samples the efficiency improvement would be smaller. We chose 10,000 samples as a good representative of a reasonably sized dataset that one could obtain.
>
> We have clarified this point in the paper when we first introduce these numbers in the experiments.
>
> **Why are different optimization methods (Newton's method and L-BFGS) used for different parameters?**
> After testing various optimization methods, we found that they converged to the same ratings, but at different rates. Specifically, Newton’s method showed faster convergence for model-specific parameters, while L-BFGS performed better for model-agnostic parameters. Therefore, we chose to use the faster method for each parameter type to optimize the efficiency of our rating system.
>
> **What specifically makes Mixtral's performance different for the Chinese and English tasks in Table 3?**
> By "different," we mean that Mixtral’s performance on Chinese and English tasks deviates significantly from its base performance, an effect that is due to Mixtral’s relatively narrow multilingual focus compared to some other models. In particular, Mixtral’s ratings reveal a notable gap: it performs significantly better on English tasks and significantly worse on Chinese tasks than it does overall.
>
> However, the univariate baseline overlooks these task-specific variances, assigning a fixed rating to Mixtral across all tasks. In this case, it artificially lowers Mixtral’s rating (and therefore all model ratings) by around 25 points for English tasks while raising it by approximately 37 points for Chinese tasks. This shift causes inaccuracies in model comparison across tasks. In contrast, Polyrating accounts for such variability, preserving Mixtral’s true performance differences across tasks. Without Polyrating, these nuanced performance differences would be masked, limiting the interpretability of Mixtral’s ratings in both Chinese and English contexts.

---

> > ### Author Response · Authors · 2024-11-18
> >
> > **Could you include additional model comparisons in Table 3 to enhance the validity of the comparisons?**
> > Yes, the full results for all 114 models are already included in Appendix Tables 7 and 8. To balance detail with readability, we included a representative subset of well-known models in Table 3, selecting models that illustrate general trends in task-specific performance. These extended tables support our findings and demonstrate how Polyrating aligns better with expected model performance across tasks compared to the univariate model.
> >
> > For instance, examining the Chinese task rating, we see that 97 of 114 models receive lower ratings in the univariate baseline compared to their base ratings. This result highlights a notable bias in its Chinese task ratings. In contrast, Polyrating assigns ratings that better reflect the actual performance differences across models, with only 67 of the 114 models showing a decreased rating for the Chinese task. More importantly, the only models to gain significant rating points are those trained by Chinese model providers, Yi, Qwen, and GLM. Similarly, for English ratings, the tables show an opposite trend: the univariate baseline improves the ratings of only 11 models, while Polyrating identifies 77 models with improved English ratings. Furthermore, Polyrating clearly shows a compression of ratings for the English tasks, as the best models generally lose rating points, while the worst models gain rating points. This is expected since most older and worse models were only (or mostly) trained on English data. In contrast, it would be much harder to conclude this from the univariate ratings.
> >
> > We have included these points in Section 4.3 as we agree with the reviewer that an analysis of the general trends is much more convincing than the specific remarks made before.
> >
> > We hope that this addresses all the reviewers' questions and concerns. We are happy to discuss any further questions the reviewer might have.

---

> > > ### Comment · Reviewer_1REa · 2024-11-20
> > > **Thank you**
> > >
> > > By checking the revisions and reading the author's responses, I think my concerns have been addressed. I believe the current score is a reasonable choice.

---

> > > > ### Author Response · Authors · 2024-11-27
> > > >
> > > > We thank the reviewer for their reply and for acknowledging that our revisions have addressed their concerns. We would greatly appreciate it if the reviewer could share any remaining reservations or areas of improvement that might still be holding back a higher score. We hope to be able to clarify and address any further questions and concerns.

---

### Official Review · Reviewer_iGCp · 2024-11-04

**Soundness:** 4
**Presentation:** 3
**Contribution:** 4
**Rating:** 8
**Confidence:** 4

**Summary:**

This paper introduces Polyrating, a novel multivariate rating system designed specifically for evaluating Large Language Models (LLMs). The system addresses three key limitations of current rating approaches: bias awareness, cost efficiency and cross-task comparability. Polyrating uses maximum a posteriori estimation with shared parameters to quantify and account for various biases in human and LLM-based evaluations. The system demonstrates up to 77% cost reduction for new tasks and 41% for new models by leveraging existing benchmark scores. It also enables direct rating comparisons across different tasks by solving the shift-invariance problem inherent in traditional rating systems. The paper provides thorough empirical validation using public datasets like Chatbot Arena and theoretical proofs of the system's optimality.

**Strengths:**

1. The paper makes good practical contributions by introducing a novel framework for LLM evaluation. Fundamental limitations in current approaches are well addressed, providing strong theoretical foundations with proper convergence guarantees and demonstrating substantial practical improvements.
2. The authors present comprehensive empirical validation across multiple datasets with comparison against baseline approaches and clear ablation studies showing the contribution of each component.
3. The significant practical impact demonstrated (up to 77% cost reduction for new tasks and 41% for new models by leveraging existing benchmark scores) offers immediate positive value to the field.
4. The paper is generally well written with clear problem motivation and logical flow from theory to implementation to results.

**Weaknesses:**

1. There is limited discussion of computational complexity and scalability analysis. The paper could include runtime analysis for both training and inference phases of Polyrating, and recommended batch sizes and optimization strategies for large-scale deployment. Similarly, a short discussion of how the MAP optimization scales with number of models (k), tasks (d) and samples (n) would be useful.

2. The hyperparameter selection process for standard deviations sigma_j and sigma_j' could be better explained. The relationship between dataset size and optimal prior values should be analyzed and sensitivity analysis showing how different prior choices affect final ratings would also be valuable.

3. The system assumes static model capabilities but the paper should address rapid model changes with, eg: analysis of rating stability over time, methods for updating ratings incrementally as new data arrives, guidelines for maintaining rating consistency across model versions, etc.

4. The paper doesn't explore well on how the system performs with extremely sparse preference data (cold-start scenarios). The authors could analyze the minimum number of preferences needed per task/model pair. Also, methods for handling completely missing task-model combinations should be detailed.

**Questions:**

1. What is the computational complexity of the optimization process and how does it scale with the number of models and tasks?
2. How sensitive is the system to the choice of priors? Could you provide guidelines for selecting appropriate priors?
3. How does the system handle rapid changes in model capabilities over time?
4. How does the system handle extremely sparse preference data for new tasks or models?

---

> ### Author Response · Authors · 2024-11-18
>
> We thank the reviewer for their detailed review and insightful questions. We are happy to hear that they consider our paper well-written and a good practical contribution and that our experiments are comprehensive and show significant improvements. Below, we address their remaining questions.
>
> **What is the computational complexity of the optimization process and how does it scale with the number of models and tasks?**
> The optimization process for Polyrating requires no more than 6 hours on a single CPU to handle a substantial dataset consisting of one million samples, 100 models, and ten tasks. Consequently, the computational cost of running Polyrating is negligible when compared to the expense of acquiring such a dataset. Moreover, the process is easily executable on commodity hardware, making it highly accessible.
>
> We also experimented with varying the number of models and games but did not observe a clear trend in computational complexity. This lack of a clear trend can be attributed to several implemented optimizations, such as grouping games with identical features into a single term within the optimization process, but also due to varying number of iterations of the optimization process that were required to converge to the optimal solution.
>
> **How sensitive is the system to the choice of priors? Could you provide guidelines for selecting appropriate priors?**
> Yes, the choice of priors, specifically the variances associated with those priors, is crucial for the performance of Polyrating. Low variance introduces heavy regularization, potentially slowing convergence and limiting adaptation to new tasks, while high variance may reduce the regularizing effect of priors, making them less impactful. Fortunately, these parameters are optimized automatically via cross-validation on the training dataset, allowing us to identify priors that best support predictive accuracy. Empirically, we found that optimal standard deviations, identified via this process, ranged from 1 to 100. Further,  based on our experiments in Figure 1 and Figure 3, these variances adjust dynamically based on dataset size: with fewer samples, lower standard deviations are typically optimal, and as sample count increases, the optimal standard deviations also tend to increase. This adaptive approach reduces manual tuning effort and improves robustness across varying data conditions. Thus, one should always use this cross-validation-based approach to find the appropriate hyperparameters. If this is not a possibility for some reason, or to obtain initial results, standard deviations around 50 provide a good default value. We note that for all our experiments, we only optimized over 10 possible values of the variances.
>
> **How does the system handle rapid changes in model capabilities over time?**
> We thank the reviewer for raising this interesting point and have now added an experiment addressing changing model capabilities, specifically model versioning, in Appendix C. As discussed there, Polyrating can incorporate model updates by imposing an extra regularization condition on the base rating of two model versions, similar to the regularization between a model’s base rating and its task-specific rating. This small adjustment enables Polyrating to converge 38% faster when obtaining 10,000 samples for new model versions. Notably, the experiment performed in the appendix only considers publicly available versions. If a model developer is evaluating small, incremental updates to their model, this convergence rate is likely to be much higher as these small updates are much more likely to lead to very similar ratings. Unfortunately, we do not have access to a (large) dataset of preferences that keep track of model performance across many different internal versions and can therefore not evaluate Polyrating in this setting.
>
> **How would Polyrating perform with extremely sparse preference data?**
> Polyrating manages sparse preference data through a Bayesian approach that incorporates priors and deviations. For models with limited task-specific data, the system reflects this uncertainty in the associated rating, clearly signaling to users that fewer observations inform the rating. In extreme cases with no data for a particular task-model pair, Polyrating initializes the rating based on the general prior, indicating an expected rating equal to the base rating but with high uncertainty. This is demonstrated in Table 2 and Table 7, where models with fewer samples, such as Claude-3.5-Sonnet, show higher variance in rating estimates. This mechanism not only ensures robust initial estimates in low-data scenarios but also communicates the confidence level associated with each rating, enhancing interpretability for users.
>
> We hope that this addresses all the reviewers' questions and concerns. We are happy to discuss any further questions the reviewer might have.

---

### Official Review · Reviewer_2PJr · 2024-11-04

**Soundness:** 3
**Presentation:** 3
**Contribution:** 2
**Rating:** 6
**Confidence:** 2

**Summary:**

The paper addressed that current rating systems for LLMs suffer from several limitations: biases, human annotations, and shift-invariance.\
Then, the authors propose POLYRATING, a multivariate rating system based on MAP estimation.\
POLYRATING uses maximum a posteriori estimation (MAP) to estimate model ratings across tasks, including base ratings and task-specific adjustments.\
It also incorporates shared parameters to model factors like answer length and readability that affect preferences.

**Strengths:**

1. The notations and equations throughout the paper are clear, well-formulated, and add rigor to the methodology.
2. The justification for POLYRATING is robust, with a well-explained rationale for the multivariate approach.
3. Extensive experiments demonstrate the effectiveness of POLYRATING, showing its superiority over traditional rating methods in several key areas.

**Weaknesses:**

1. The modeling features are somewhat ambiguous; it is unclear which specific features are applied for each task, or the criteria used for feature selection. More detailed information on feature selection per task would be helpful.
2. POLYRATING employs both model-shared and model-specific weights. It would be useful to clarify how the system initializes weights for newly added models and whether it can efficiently update ratings for new models without requiring a full retraining of the rating system.

**Questions:**

Please refer to Weaknesses.

---

> ### Author Response · Authors · 2024-11-18
>
> We thank the reviewer for their detailed review and insightful questions. We are pleased to hear that they find our notation clear and rigorous, our justification for Polyrating robust, and appreciate our extensive experiments that demonstrate its effectiveness. Below, we address the remaining questions and aim to clarify any outstanding points.
>
> **Could you provide more detailed information on the feature selection per task?**
> Certainly. In this response, we discuss this for each experiment separately.
>
> **Bias Detection.** In our bias detection experiments, we started by identifying biases that could exist in judges and that we would like to measure. We then map each bias conceptually into a function that quantifies it within an answer. For example, for readability, we use the Flesch Reading Ease score to estimate the influence of readability on a judge’s preference. The functional form of each of these biases is given in Table 4 on page 20. Once the functional form is specified, Polyrating can be applied to measure the influence of the bias.
>
> **Sample Efficiency.** In Section 4.2, we specify the applied rating modeling formula in each experiment. In all these experiments, feature selection serves to establish meaningful priors across tasks. For example, the use of $R_\text{base}^m + \beta_\text{task}^m [[ g \in D_\text{task} ]]$ with a prior on  $\beta_\text{task}^m$ ensures that a model's performance on a task and its base performance is reasonably close, which supports improved convergence. The only exception to this approach is the inclusion of log length as a feature in LLM-based judges. This feature counteracts the well-known increased length bias in LLM judgments compared to human judgments, aligning the priors more effectively.
>
> **Multivariate Leaderboard.** In our final experiment, we add a feature per task to measure performance directly. These task-specific features are designed as described in L191-199, using $\beta_\text{task}^m [[ g \in D_\text{task} ]]$ to capture the performance on each task.
>
> **Could you clarify how Polyrating initializes weight for newly added models? Can Polyrating efficiently update ratings for new models without retraining the rating system?**
> Polyrating is a statistical method that initializes ratings for new models through priors. For base ratings, this manifests as a uniform prior across all ratings, while for features we employ a normal prior centered at zero. As new preferences are added, the system automatically incorporates these priors when optimizing the loss function. Thus, Polyrating does not require retraining and can seamlessly add new models and preferences on the fly. One would simply continue the optimization process from where it left off, but now with the incorporation of the new models or preferences. Even if full retraining were necessary, the process would take under six hours on a single CPU core, which is negligible compared to the total time required to obtain a significant number of samples that require LLM inference on expensive GPUs and human judgments.
>
> We hope that this addresses all the reviewers' questions and concerns. We are happy to discuss any further questions the reviewer might have.

---

> > ### Comment · Reviewer_2PJr · 2024-11-21
> >
> > Thank you for your response.\
> > I will keep my positive score.

---

> > > ### Author Response · Authors · 2024-11-27
> > >
> > > We thank the reviewer for their reply. We would greatly appreciate it if the reviewer could share any remaining reservations or areas of improvement that might still be holding back a higher score. We hope to be able to clarify and address any further questions and concerns.

---

### Official Review · Reviewer_6vXX · 2024-11-04

**Soundness:** 3
**Presentation:** 3
**Contribution:** 3
**Rating:** 6
**Confidence:** 3

**Summary:**

The authors propose, POLYRATING, an large language model evaluation system. POLYRATING can detect and quantify biases that affect human preferences, improve sample efficiency, reduce evaluation costs, and enable cross-task model scoring comparisons. It achieves a significant improvement on the sample efficiency, reducing the cost of human evaluation by up to 77% for new tasks and 41% for new models.

**Strengths:**

- POLYRATING is a multivariate rating system that can detect and quantify biases affecting human and LLM-based evaluations of LLMs.
- Such effective system reduces the cost of human evaluation by up to 77% for new tasks and 41% for new models, which is meaningful and practical in the real-world applications.
- The paper is easy to follow.
- Comprehensive experiments are conducted, demonstrating the effectiveness of the proposed method.

**Weaknesses:**

Null

**Questions:**

Will POLYRATING also be appliable to other domains that use large language models, such as computer vision, information retrieval, etc.? How is the cost for the adaptation to these domains?

---

> ### Author Response · Authors · 2024-11-18
>
> We thank the reviewer for their detailed review and insightful questions. We are pleased that they found our paper easy to follow, Polyrating effective with meaningful real-world applications, and appreciated our comprehensive experiments. We will address their question regarding applicability across domains in our response. If the reviewer has further concerns or suggestions for improvement, we would be happy to consider them. Otherwise, we would appreciate consideration for an improved score.
>
> **Will Polyrating also be applicable to domains other than large language models such as computer vision, information retrieval, etc.? What is the cost of the adaptation to these domains?**
> Yes, Polyrating is applicable to other domains, including multimodal models and domain-specific tasks like information retrieval, as long as preference data (outcomes and judgments) is available. We have also successfully applied Polyrating beyond AI evaluation, such as in chess.
>
> To adapt Polyrating to new domains, only minor adjustments are needed:
> - **Data Availability:** We used open-source preference datasets for LLMs in our work. For other domains (e.g., multimodal), open-source datasets on the same scale do not yet exist, though Polyrating requires no modifications. Thus, creators of such preference datasets for their specific application can directly apply Polyrating.
> - **Custom Factors:** Domain-specific factors may need to be defined (e.g., type of visual content in multimodal models). These factors depend on the specific use case for which Polyrating is applied. However, the manual work involved in this feature specification is limited to negligible.
>
> These simple adaptations allow Polyrating to retain its core benefits—improving sample efficiency and reducing evaluation costs—across various applications.
>
> We hope that this addresses all the reviewers' questions and concerns. We are happy to discuss any further questions the reviewer might have.

---

### Author Response · Authors · 2024-11-18

$\newcommand{R}{\textcolor{green}{6vXX}}$
$\newcommand{S}{\textcolor{blue}{2PJr}}$
$\newcommand{T}{\textcolor{purple}{iGCp}}$
$\newcommand{U}{\textcolor{purple}{1REa}}$
We thank the reviewers for their high-quality reviews and insightful questions. We are particularly happy to hear they consider our paper well-written and easy to follow ($\R, \S, \T$), that the justification for Polyrating is strong ($\S, \U$), and that it is effective ($\R, \S, \T$). We could not identify any common questions between the reviewers, and will only include a comprehensive overview of the changes made to the paper in response to the reviews.

**Experiment on Model Versioning (Appendix C)**
We included an experiment showcasing that Polyrating can also improve convergence of the rating of new model versions in Appendix C. For developers who create multiple versions of a model simultaneously or in quick succession, it is essential to evaluate the relative performance of these versions as cost-effectively as possible. However, this process can be computationally expensive, especially when human evaluations are used.  By applying an extra prior between the ratings of two model versions, one can easily extend Polyrating to improve sample efficiency for new model versions. Specifically, we converge 38% faster when obtaining 10,000 samples for new model versions.

**Clarification Exclusion Multivariate Rating Systems (L438-442,L446-452)**
In the revised version, we explain more thoroughly in our Related Work section why prior works on multivariate rating systems are inapplicable to the experiments performed in our paper.

**Expanding the Discussion in Section 4.3 (L405-420)**
We expanded the discussion in Section 4.3 to more clearly refer to overall trends, which more convincingly show the benefits of Polyrating. For instance, the only models that gain significant rating points for the Chinese task are those trained by Chinese model providers. In contrast, the univariate approach shows no such pattern.

**Clarification Efficiency Metric (L320)**
We clarified how we computed the efficiency improvements for Polyrating when we first introduced them in the experiments. Specifically, in L320 we now indicate that the reported number is the fraction of extra samples the univariate baseline requires to obtain the same loss as Polyrating.

**Inclusion Glicko in Figure 4 (L803-806)**
We include Glicko in Figure 4 to show that approximate systems are ineffective when used for language models, and perform much worse than the exact systems based on the BT-Model used by Polyrating.

---

### Author Response · Authors · 2024-12-04

We sincerely thank all four reviewers for their constructive comments and questions, which have significantly helped us improve our work. We are particularly encouraged by the reviewers' recognition of our paper's strengths, as summarized below:
$\newcommand{R}{\textcolor{green}{6vXX}}$
$\newcommand{S}{\textcolor{blue}{2PJr}}$
$\newcommand{T}{\textcolor{purple}{iGCp}}$
$\newcommand{U}{\textcolor{orange}{1REa}}$

**Significant Improvement**
- The presented experiments rigorously demonstrate the effectiveness of Polyrating ($\S,\R,\T$).
- The measured improvement over prior work is substantial and practical in real-world applications ($\R,\S,\T$).

**Good Presentation**
- The paper is well written with a clear logical flow ($\T,\S,\R,\U$).
- The limitations of prior approaches are clearly identified and directly addressed by Polyrating ($\U,\T,\S$).

**Theoretical Guarantees**
- The theoretical analysis of Polyrating provides strong theoretical foundations ($\U,\T$).

We acknowledge that our initial submission had areas for improvement. In response to the reviewers’ thorough reviews, we have provided the following additional information and experiments which we incorporated in the revision of the paper:

- **Model Versioning**: We showcased that Polyrating can be used to improve the convergence rate by 38% for different versions of the same model.
- **Exclusion Multivariate Rating Systems**: We clarified that existing multivariate rating systems are inapplicable to the settings evaluated in our experiments.
- **Improved Result Discussion**: We expanded the discussion in Section 4.3 to more clearly refer to overall trends, which more convincingly show the benefits of Polyrating.
- **Clarification Efficiency**: We clarified the way in which we compute efficiency improvements.

We additionally clarified the following significant points to individual reviewers:
- **Applications Beyond LLMs**: Polyrating can be applied to areas outside of LLMs, as long as preference datasets are available.
- **No Retraining Needed and Limited Computational Complexity**: New models can be added on the go, and the maximum time spent for a single run was 6h on a single CPU core.

Once again, we deeply appreciate the valuable feedback and guidance provided by the reviewers.

Best regards,
The authors

---

### Meta-Review · Area_Chair_M11t · 2024-12-21

**Metareview:**

This paper focuses on improving the performance scoring system for LLMs and proposes a new rating system called Polyrating based on maximum a posteriori estimation. All four reviewers agreed that this paper exceeds the ICLR 2024 acceptance threshold, with a clear presentation and theoretical support. However, reviewers also pointed out some further weaknesses, including: 1) more discussion should be conducted on the efficiency of the proposed method and its use cases; 2) A clearer description of the method's details is necessary. During the rebuttal phase, the authors addressed most of these issues, and it is believed that these concerns will be resolved and reflected in the final version.

**Additional Comments On Reviewer Discussion:**

In the rebuttal phase, the authors responded to all the reviewers' questions, further helping to resolve their issues.

---

### Decision · Program_Chairs · 2025-01-22

Accept (Poster)